# IntraMix: Intra-Class Mixup Generation for Accurate Labels and Neighbors

**Shenghe Zheng, Hongzhi Wang,**[*] **Xianglong Liu**

Massive Data Computing Lab, Harbin Institute of Technology

shenghez.zheng@gmail.com    wangzh@hit.edu.cn

## Abstract

Graph Neural Networks (GNNs) have shown great performance in various tasks, with the core idea of learning from data labels and aggregating messages within the neighborhood of nodes. However, the common challenges in graphs are twofold: insufficient accurate (high-quality) labels and limited neighbors for nodes, resulting in weak GNNs. Existing graph augmentation methods typically address only one of these challenges, often adding training costs or relying on oversimplified or knowledge-intensive strategies, limiting their generalization. To simultaneously address both challenges faced by graphs in a generalized way, we propose an elegant method called IntraMix. Considering the incompatibility of vanilla Mixup with the complex topology of graphs, IntraMix innovatively employs Mixup among inaccurate labeled data of the same class, generating high-quality labeled data at minimal cost. Additionally, it finds data with high confidence of being clustered into the same group as the generated data to serve as their neighbors, thereby enriching the neighborhoods of graphs. IntraMix efficiently tackles both issues faced by graphs and challenges the prior notion of the limited effectiveness of Mixup in node classification. IntraMix is a theoretically grounded plug-in-play method that can be readily applied to all GNNs. Extensive experiments demonstrate the effectiveness of IntraMix across various GNNs and datasets. Our code is available at: https://github.com/Zhengsh123/IntraMix.

## 1 Introduction

Graph Neural Networks (GNNs) have demonstrated great potential in various tasks [52]. However, most graph datasets lack high-quality labeled data and node neighbors, underscoring two key challenges for GNNs: the dual demands for accurate labels and rich neighborhoods [5].

Data augmentation is one way to address these two issues, but research on graph data augmentation is insufficient. Additionally, it is challenging to apply widely studied data augmentation ways designed for Euclidean data such as images to graphs due to their non-Euclidean nature [15]. Therefore, unique graph augmentation methods are needed. While graph augmentation aims to generate high-quality labeled data and enrich node neighbors, most methods either focus on one aspect and often suffer from poor generalization ability. For example, some require generators, incurring additional training costs [51, 24]. Others rely on overly simplistic ways such as random drop, yielding little improvements [7]. Still, some depend on excessive prior knowledge, weakening generalization abilities [48]. Therefore, there is an urgent need for an effective and generalized augmentation method that can produce high-quality labeled nodes and adequately enrich node neighbors.

Reviewing existing methods, we find that they overlook the potential of low-quality labeled data, which can be obtained at a low cost. Extracting information from such data could enrich data diversity.

---

[*]Corresponding author.

38th Conference on Neural Information Processing Systems (NeurIPS 2024).

The cause of low-quality labels is the noise in labels, which results in a distribution different from the real one [10]. The noise direction is usually mixed, so a natural idea is to blend noisy data, leveraging noise directionality to neutralize it and produce accurate labels. Therefore, Mixup [50] comes into our eyes as a method that involves mixing data. It is defined as $\hat{x} = \lambda x_i + (1-\lambda)x_j, \hat{y} = \lambda y_i + (1-\lambda)y_j$, where $(x_i, y_i), (x_j, y_j)$ represent selected data, and $y$ represents the label. Despite Mixup excels in Euclidean data, experiments suggest its limited ability in node classification task [45]. Therefore, a question emerges: **Can Mixup solve augmentation problems for node classification?**

Due to the characteristics of graphs, using Mixup is challenging. The main reason Mixup performs poorly in node classification can be attributed to the graph topology. In Euclidean data such as images, the data generated by Mixup are independent [23], whereas in graphs, the generated nodes need to be connected to other nodes to be effective. This means that in graphs, the data generated by vanilla Mixup is difficult to determine its neighbors and connecting them to any class of nodes is inappropriate, as its distribution does not belong to any current class of data [45]. Hence, the complex topology of the graph directly leads to the inapplicability of vanilla Mixup.

To address the issues, we propose IntraMix, a novel augmentation method for node classification, as shown in Figure 1(b). The core idea is that since nodes generated by Mixup between different class data are hard to find neighbors for, we mix nodes within the same class obtaining by pseudo-labeling (low-quality labels) instead [21]. Hereby, the main benefit of this approach is that it addresses the primary challenge about neighborhood of applying Mixup on graphs. The generated node features are no longer a mixture of features from multiple groups, making it easier to find neighbors based on their respective groups. Additionally, the generated nodes have higher quality labels. Intuitively, if we simplify the label noise as $\epsilon \sim N(0, \sigma^2)$, the mean distribution of two noises $\bar{\epsilon} \sim N(0, \frac{1}{2}\sigma^2)$, with a smaller variance, increases the likelihood that the generated label is accurate. Therefore, we address the sparse high-quality labels by Intra-Class Mixup. It is important to emphasize that although some works have used Intra-Class Mixup [25], we are the first to use it as the sole augmentation method and to analyze it in depth.

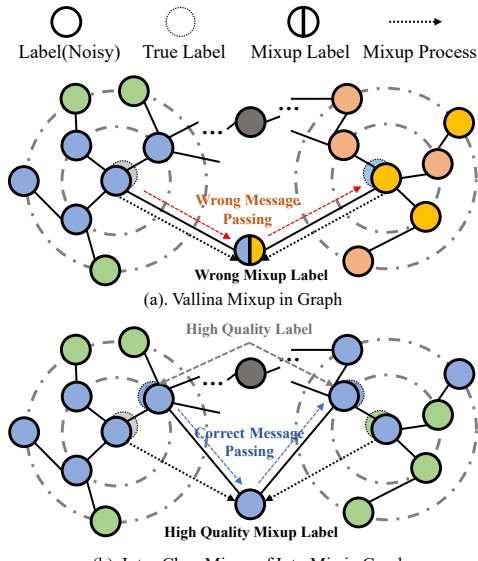

Figure 1: a). Vanilla Mixup may retain label noise, and connecting generated nodes to original nodes may lead to incorrect propagation. b). IntraMix generates high-quality data by Intra-Class Mixup and enriches neighborhoods while preserving correctness by connecting generated data to high-quality nodes.

Once the method for generating data is determined, the neighborhood selection method becomes very straightforward. For the neighbor selection strategy of the generated node $v$, we connect $v$ to two nodes with high confidence of the same class with $v$. This has two benefits. Firstly, based on the assumption that nodes of the same class are more likely to appear as neighbors (neighborhood assumption) [53], we reasonably find neighbors for $v$, providing it with information gain in training and inferencing. Secondly, by connecting $v$ to two nodes that belong to the same class, we not only bring message interaction to the neighbors of these two nodes but also reduce the impact of noise that may still be present in direct connecting high-quality labels. In this neighbor selection way, we construct rich and reasonable neighborhood for nodes, enhancing the knowledge on the graph, which allows for the development of stronger GNN models for downstream tasks.

Therefore, IntraMix simultaneously addresses two challenging issues faced by graphs and exhibits strong generalization capabilities in an elegant way. Our key contributions are as follows:

• For the first time, we introduce Intra-Class Mixup as the core data augmentation in node classification, highlighting its effectiveness in generating high-quality labeled data.

• The proposed IntraMix tackles sparse labels and incomplete neighborhoods faced by graph datasets through an elegant and generalized way of Intra-Class Mixup and neighborhood selection.

• Extensive experiments demonstrate that IntraMix improves the performance of GNNs on diverse datasets. Theoretical analysis elucidates the rationale behind IntraMix.

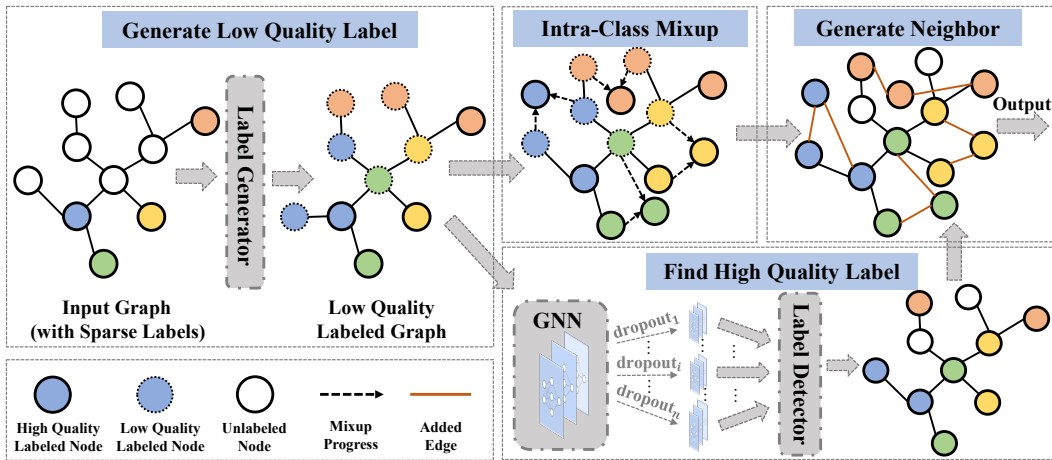

Figure 2: The workflow of IntraMix involves three main steps. First, it utilizes pseudo-labeling to generate low-quality labels for unlabeled nodes. Following that, Intra-Class Mixup is employed to generate high-quality labeled nodes from low-quality ones. Additionally, it identifies nodes with high confidence in the same class and connects them, thus constructing a rich and reasonable neighborhood.

## 2 Preliminaries

**Notations:** Given a graph $G = (V, E)$, where $V = \{v_i\}_{i=1}^{N}$ is the node set, and $E$ represents the edge set, the adjacency relationship between nodes can be represented by $A \in \{0, 1\}^{N \times N}$, where $A_{ij} = 1$ if and only if $(v_i, v_j) \in E$. We use $X \in R^{N \times D}$ to denote the node feature. The node labels are represented by $Y$. Based on the presence or absence of labels, $V$ can be divided into $D_l = \{(x_{l_1}, y_{l_1}), ...(x_{l_N}, y_{l_N})\}$ and $D_u = \{x_{u_1}, ...x_{u_N}\}$. We can use pseudo-labeling to assign low-quality labels $Y_u$ to nodes in $D_u$, getting a low-quality set $D_p = \{(x_{u_1}, y_{u_1}), ...(x_{u_N}, y_{u_N})\}$. $N_i = \{v_j | A_{ij} = 1\}$ are the neighbors of $v_i$. Detailed notation list can be found in Appendix A.

**Node Classification with GNNs:** Given a graph $G$, the node classification involves determining the category of nodes on $G$. GNNs achieve this by propagating messages on $G$, representing each node as a vector $h_v$. The propagation for the $k$-th layer of a GNN is represented as follows:

$$h_v^k = COM(h_v^{k-1}, AGG(\{h_u^{k-1} | u \in N_v\})) \tag{1}$$

where *COM* and *AGG* are COMBINE and AGGREGATE functions, respectively, and $h_v^k$ denotes the feature of $v$ at the $k$-th layer. The output $h_v$ in the last layer of GNN is used for classification as $y_v = softmax(h_v)$, where $y_v$ is the predicted label for $v$.

## 3 Methodology

In this section, we provide a detailed explanation of IntraMix. Firstly, we present the Intra-Class Mixup in 3.1. It generates high-quality labeled nodes from low-quality data, addressing the issue of label sparsity. Then, we show the method for finding node neighbors in 3.2. Next, in 3.3, we conclude the workflow and conduct complexity analysis in 3.4. The framework is shown in Figure 2.

### 3.1 Intra-Class Mixup

**Motivation:** In supervised learning, having more labels typically allows for learning finer classification boundaries and getting better result [39]. However, obtaining accurately labeled data is costly in node classification. Nevertheless, directly utilizing low-quality labels from pseudo-labeling introduces noise detrimental to learning. As we know, low-quality pseudo-labeled data are often closer to the boundaries that models can learn from current data, containing useful information for classification [21]. It is possible to generate high-quality data from them as mentioned in Sec 1. Due to the fact that label noise is typically spread in multiple directions, we take full advantage of the directional nature of the noise. By blending data, we make the noise distribution closer to zero,

thereby reducing the noise in the labels. At the same time, considering that data generated by vallina Mixup is hard to find neighbors, we innovatively propose Intra-Class Mixup. It generates data with only a single label, making it easy to determine the neighborhood. It not only generates high-quality data but also facilitates the finding of neighbors. Additionally, we can provide theoretical guarantees.

**Approach**: We use pseudo-labeling to transform the unlabeled nodes $D_u$ into nodes with low-quality labels $D_p$. Then, we get $D = D_l \cup D_p$, where there are a few high-quality and many low-quality labels. In contrast to the vanilla Mixup, which is performed between random samples, we perform Mixup among nodes with the same low-quality labels to obtain high-quality labeled data guaranteed by Theorem 3.1. The generated dataset is represented as:

$$D_m = \{(\hat{x}, \hat{y}) | \hat{x} = M_\lambda(x_i, x_j), \hat{y} = y_i = y_j\}, \tag{2}$$

where

$$M_\lambda(x_i, x_j) = \lambda x_i + (1 - \lambda)x_j, (x_i, y_i), (x_j, y_j) \in D. \tag{3}$$

The number of generated nodes is manually set. The labels in $D_m$ are of higher quality compared to $D$, a guarantee provided by Theorem 3.1. The proof can be found in Appendix B.1. In other words, the generated labels exhibit less noise than those of their source nodes. Through Intra-Class Mixup, we obtain a dataset with high-quality labels, leading to improved performance of GNNs.

**Assumption 3.1.** *Different classes of data have varying noise levels, i.e., $P_{\text{noise}}(y_i|x) = P(y_i|x) + \epsilon_i$, where $P_{\text{noise}}(y_i|x)$ and $P(y_i|x)$ represent the label distribution of class $i$ with and without noise, and $\epsilon_i \sim N(0, \sigma_i^2)$ represent the noise.*

**Theorem 3.1.** *For Intra-Class Mixup satisfying Equation 2 and Assumption 3.1, $P(\hat{\epsilon} < \epsilon) = \frac{2}{\pi} \arctan((\lambda^2 + (1 - \lambda)^2)^{-\frac{1}{2}})$, and $\frac{E(\hat{\epsilon})}{E(\epsilon)} = [\lambda^2 + (1 - \lambda)^2]^{\frac{1}{2}} < 1$, where $E(\cdot)$ represents the expectation, and $\hat{\epsilon}$ and $\epsilon$ denote the noise in the generated data and the original data, respectively.*

## 3.2 Neighbor Selection

**Motivation:** The strength of GNN lies in gathering information from neighborhoods to mine node features [14]. After generating node $v$ in Sec 3.1, to leverage the advantages of GNN, it is necessary to find neighbors for $v$. We aim to construct a neighborhood that satisfies two requirements: a). The neighborhood is suitable for $v$; b). The neighbors of $v$ can obtain richer information through $v$. If $v$ is simply connected to nodes that generated it, as the nodes used for Mixup mostly have low-quality labels, it is prone to resulting in incorrect information propagation. Since nodes of the same class are more likely to appear in the neighborhood in homogeneous graphs, a natural idea is to connect $v$ with nodes of high confidence in the same class. In this way, we can find the correct neighbors for $v$ and, acting as a bridge, connect the neighbors of two nodes of the same class through $v$ to obtain more information from the graph, as shown in Figure 1(b).

**Approach:** As mentioned above, neighborhood selection involves two steps. First, finding nodes highly likely to be of the same class as $v$, and second, determining how to connect $v$ with these nodes. We will now introduce them separately in the following.

Finding nodes with a high probability of belonging to the same class can be transformed into the problem of finding high-quality labeled data. Then, we ingeniously design an ensemble method without extra training costs. We employ the GNN for pseudo-labeling to predict the label of nodes under $n$ dropout probabilities. Nodes consistently predicted in all $n$ trials are considered high-quality. This is an ensemble method with $n$ GNNs but without $n$ training costs, significantly reducing cost. Details are in Appendix D.3. It is expressed as:

$$D_h = \{(x, y) | f_1(x) = ... = f_n(x), (x, y) \in D\}, \tag{4}$$

where $f_i$ represents GNNs with different dropout probabilities. After obtaining the high-quality set $D_h$, it is time to establish neighborhoods between $D_h$ and $D_m$ generated by Mixup. To ensure the correctness of neighbors, we adopt the way of randomly connecting the generated data to high-quality nodes of the same class. The augmented edge set $\hat{E}$ of the original set $E$ can be expressed as:

$$\hat{E} = E \cup \{e(\hat{x}, x_i) | (\hat{x}, y) \in D_m, (x_i, y) \in D_h\}, \tag{5}$$

where $e(a, b)$ represents an edge between nodes $a$ and $b$. In this way, we not only find reasonable neighbors for the generated nodes but also establish an information exchange path between two nodes

---

**Algorithm 1** Workflow of IntraMix

---

**Input:** Graph $G = (V, E)$, $V$ can be divided into $D_l$ and $D_u$, Class of nodes $C$, GNN model $f$

1: Generate pseudo labels for $D_u$ using $f$, get $\hat{D}_u$
2: $D = D_l \cup \hat{D}_u$
3: Generate Mixup set $D_m = \{V_m, E_m\}$ as Equation.2
4: $V = V \cup V_m$
5: Generate high-quality set $D_h$ according to Equation.4
6: **for** $(\hat{x}, \hat{y}) \in D_m$ **do**
7:     $E \cup \{e(\hat{x}, x_i), e(\hat{x}, x_j)\}$, where $(x_i/x_j, \hat{y}) \in D_h$
8: **end for**

**Output:** the augmented graph $G = (V, E)$

---

of the same class. Additionally, by not directly connecting the two nodes, potential noise impacts are avoided. The elimination effect of noise is guaranteed by Theorem 3.2. The detailed proof can be found in Appendix B.2. Through this method, the issue of missing neighborhoods in the graph is alleviated, and a graph with richer structural information is constructed for downstream tasks.

**Assumption 3.2.** *The label noise can be represented as node noise, i.e.,* $P_{noise}(x|y_i) = P(x|y_i) + \delta_i$, *where* $\delta_i \sim N(0, \sigma_{xi}^2)$. *Equation* 1 *simplifies to* $h_v^k = MLP^k[(1 + \eta_k)h_v^{k-1} + \frac{1}{|N_v|}\sum_{u \in N_v} h_u^{k-1}]$, *where* $\eta_k$ *is learnable.* $m$ *and* $n$ *are nodes from the* $i$-*th class,* $x_m \sim P(x|y_i)$, $x_n \sim P_{noise}(x|y_i)$.

**Theorem 3.2.** *In a two-layer GNN, we have* $\frac{E(\hat{\delta})}{E(\delta)} = \sqrt{(\lambda^2 + (1-\lambda)^2) + \frac{1}{4(2+\eta_1+\eta_2)}}$, *where* $\delta$ *represents the noise impact directly connecting* $m$ *and* $n$ *and* $\hat{\delta}$ *is the impact through IntraMix.*

The $\frac{E(\hat{\delta})}{E(\delta)}$ in Theorem 3.2 can be controlled to be less than 1 by adjusting the learnable $\eta$, indicating that the neighbor selection method of IntraMix leads to a smaller noise impact. Therefore, Theorem 3.2 guarantees that the neighborhood selection strategy proposed by IntraMix can increase the richness of information in the graph while reducing the impact of noise.

### 3.3 Workflow

In this section, we introduce the overall workflow of IntraMix. For details, please refer to Algorithm 1. The data augmentation begins by generating low-quality labels for unlabeled nodes through pseudo-labeling (lines 1). Following that, high-quality labeled data is generated by Intra-Class Mixup. Subsequently, we use the neighborhood selection strategy to select appropriate neighbors for the generated nodes (lines 6-8). The output is a graph that is better suited for downstream tasks.

### 3.4 Complexity Analysis

Overall, the time consumption of using IntraMix can be divided into four aspects: generating pseudo-labels for nodes, generating new nodes, neighborhood discovery, and the downstream task. Next, we will analyze the time complexity of each aspect separately.

In the pseudo-label generation process, there is no additional training cost since the model used is the same GNN employed for downstream tasks. The second part of the time cost comes from the pseudo-labeling process, which only involves $kN$ inference steps, where $N$ represents the number of nodes selected for pseudo-labeling and $k$ represents the number of pseudo-labeling is performed in parallel. The time cost for this step is minimal.

Next is the process of generating new nodes. Assuming the number of generated nodes is $m$, the time cost incurred during the Mixup generation is $O(m)$. Simultaneously, the time cost for finding the neighborhood for these $m$ nodes is also $O(m)$. Next, we analyze the subsequent training time consumption of IntraMix. Assuming the original time complexity of the GNN is $O(|V| \times F \times F') + O(|E| \times F')$, where $F$ denotes the input feature dimension of nodes, and $F'$ is the hidden layer dimension of GNN. The time complexity after using IntraMix is $O(|V| \times F \times F') + O(|E| \times F') + O(m \times F \times F') + O(2m \times F') + O(m)$. As in most cases, $m \ll |V|$, the time complexity is in the same order of magnitude as the original GNN. Therefore, from an overall perspective, IntraMix does not introduce significant additional time cost to the original framework.

Table 1: Semi-supervised node classification accuracy(%) on medium-scale graphs. The average result of 30 runs is reported on five datasets.

| Models | Strategy | Cora | CiteSeer | Pubmed | CS | Physics |
|---|---|---|---|---|---|---|
| GCN | Original | 81.51 ± 0.42 | 70.30 ± 0.54 | 79.06 ± 0.31 | 91.24 ± 0.43 | 92.56 ± 1.31 |
| | GraphMix | 82.29 ± 3.71 | 74.55 ± 0.52 | 82.82 ± 0.53 | 91.90 ± 0.22 | 90.43 ± 1.76 |
| | CODA | 83.47 ± 0.48 | 73.48 ± 0.24 | 78.50 ± 0.35 | 91.01 ± 0.75 | 92.57 ± 0.41 |
| | DropMessage | 83.33 ± 0.41 | 71.83 ± 0.35 | 79.20 ± 0.25 | 91.50 ± 0.31 | 92.74 ± 0.72 |
| | MH-Aug | 84.21 ± 0.38 | 73.82 ± 0.82 | 80.51 ± 0.32 | 92.52 ± 0.37 | 92.91 ± 0.46 |
| | LA-GCN | 84.61 ± 0.57 | 74.70 ± 0.51 | 81.73 ± 0.71 | 92.60 ± 0.26 | 93.26 ± 0.43 |
| | NodeMixup | 83.47 ± 0.32 | 74.12 ± 0.35 | 81.16 ± 0.21 | 92.69 ± 0.44 | 93.97 ± 0.45 |
| | IntraMix | **85.25 ± 0.42** | **74.80 ± 0.46** | **82.98 ± 0.54** | **92.86 ± 0.04** | **94.27 ± 0.14** |
| GAT | Original | 82.04 ± 0.62 | 71.82 ± 0.83 | 78.00 ± 0.71 | 90.52 ± 0.44 | 91.97 ± 0.65 |
| | GraphMix | 82.76 ± 0.62 | 73.04 ± 0.51 | 78.82 ± 0.44 | 90.57 ± 1.03 | 92.90 ± 0.42 |
| | CODA | 83.36 ± 0.31 | 72.93 ± 0.42 | 79.37 ± 1.33 | 90.41 ± 0.41 | 92.09 ± 0.62 |
| | DropMessage | 82.20 ± 0.24 | 71.48 ± 0.37 | 78.14 ± 0.25 | 91.02 ± 0.51 | 92.03 ± 0.72 |
| | MH-Aug | 84.52 ± 0.91 | 73.44 ± 0.81 | 79.82 ± 0.55 | 91.26 ± 0.35 | 92.72 ± 0.42 |
| | LA-GAT | 84.72 ± 0.45 | 73.71 ± 0.52 | 81.04 ± 0.43 | 91.52 ± 0.31 | 93.42 ± 0.45 |
| | NodeMixup | 83.52 ± 0.31 | 74.30 ± 0.12 | 81.26 ± 0.34 | **92.69 ± 0.21** | 93.87 ± 0.30 |
| | IntraMix | **85.03 ± 0.45** | **74.50 ± 0.24** | **81.76 ± 0.32** | 92.40 ± 0.24 | **94.12 ± 0.24** |
| SAGE | Original | 78.12 ± 0.32 | 68.09 ± 0.81 | 77.30 ± 0.74 | 91.01 ± 0.93 | 93.09 ± 0.41 |
| | GraphMix | 80.09 ± 0.82 | 70.97 ± 1.21 | 79.85 ± 0.42 | 91.55 ± 0.33 | 93.25 ± 0.33 |
| | CODA | 83.55 ± 0.14 | 73.24 ± 0.24 | 79.28 ± 0.46 | 91.64 ± 0.41 | 93.42 ± 0.36 |
| | MH-Aug | 84.50 ± 0.39 | **75.25 ± 0.44** | 80.68 ± 0.36 | 92.27 ± 0.49 | 93.58 ± 0.53 |
| | LA-SAGE | 84.41 ± 0.35 | 74.16 ± 0.32 | 80.72 ± 0.42 | 92.41 ± 0.54 | 93.41 ± 0.31 |
| | NodeMixup | 81.93 ± 0.22 | 74.12 ± 0.44 | 79.97 ± 0.53 | 91.97 ± 0.24 | 94.76 ± 0.25 |
| | IntraMix | **84.72 ± 0.34** | 74.37 ± 0.45 | **81.02 ± 0.49** | **92.80 ± 0.26** | **94.87 ± 0.04** |
| APPNP | Original | 80.03 ± 0.53 | 70.30 ± 0.61 | 78.67 ± 0.24 | 91.79 ± 0.55 | 92.36 ± 0.81 |
| | GraphMix | 82.98 ± 0.42 | 70.26 ± 0.43 | 78.73 ± 0.45 | 91.53 ± 0.61 | 94.12 ± 0.14 |
| | DropMessage | 82.37 ± 0.23 | 72.65 ± 0.53 | 80.04 ± 0.42 | 91.25 ± 0.51 | 93.54 ± 0.63 |
| | MH-Aug | 85.04 ± 0.41 | 74.52 ± 0.32 | 80.71 ± 0.31 | 92.95 ± 0.34 | 94.03 ± 0.25 |
| | LA-APPNP | 85.42 ± 0.33 | 74.83 ± 0.29 | 81.41 ± 0.55 | 92.71 ± 0.47 | 94.52 ± 0.27 |
| | NodeMixup | 83.54 ± 0.45 | 75.12 ± 0.33 | 79.93 ± 0.12 | 92.82 ± 0.24 | 94.34 ± 0.22 |
| | IntraMix | **85.99 ± 0.48** | **75.25 ± 0.42** | **81.96 ± 0.34** | **93.24 ± 0.21** | **94.79 ± 0.14** |

# 4 Experiment

In this section, we show the excellent performance of IntraMix in both semi-supervised and full-supervised tasks with various GNNs across multiple datasets in Sec 4.1 and Sec 4.2. Sec 4.3 highlights the inductive learning ability of IntraMix while detailed ablation experiments for in-depth analysis are presented in Sec 4.4. Additionally, we analyze how IntraMix overcomes over-smoothing in Sec 4.5 and evaluate IntraMix on heterophilic and heterogeneous graphs in Sec 4.6 and Sec 4.7, respectively.

## 4.1 Semi-supervised Learning

**Datasets:** We evaluate IntraMix on commonly used medium-scale semi-supervised datasets for node classification, including Cora, CiteSeer, Pubmed [34], CS, and Physics [35]. We follow the original splits for these datasets. We also conduct semi-supervised experiments on large-scale graphs, including ogbn-arxiv [16] and Flickr [49]. To alter the original splits for full-supervised training on these datasets, we use 1% and 5% of the original training data for semi-supervised experiments, respectively. Details can be found in Appendix C.1.

**Baselines:** We utilize four popular GNNs: GCN [19], GAT [40], GraphSAGE (SAGE) [14], and APPNP [11]. Additionally, we compare IntraMix with various mainstream graph augmentation methods [42, 6, 7, 30, 24, 25]. Details of the baselines can be found in Appendix C.2. For each graph augmentation applied to each GNN, we use the same hyperparameters for fairness. When comparing with other methods, we use the settings from their open-source code and report the average results over 30 runs. All experiments are conducted on a single NVIDIA RTX-3090.

**Result:** It is crucial to note that semi-supervised experiments are more important than full-supervised ones. This is primarily due to the sparse labels in most real-world graphs. The semi-supervised results reflect the method's potential when applied to real-world situations. Observing the results in Table 1, it is evident that IntraMix shows superior performance across almost all GNNs and datasets.

Table 2: Semi- and full-supervised node classification accuracy on large-scale graphs. The average result of 10 runs is reported. Training size refers to the proportion of training data used for training.

| Model | Strategy | ogbn-arxiv | | | Flickr | | |
|---|---|---|---|---|---|---|---|
| Training Size | | 1% | 5% | 100% | 1% | 5% | 100% |
| GCN | Original | 62.99 ± 1.01 | 68.65 ± 0.31 | 71.74 ± 0.29 | 46.02 ± 1.25 | 48.50 ± 0.49 | 51.88 ± 0.41 |
| | FLAG | 63.68 ± 0.91 | 69.14 ± 0.43 | 72.04 ± 0.20 | 46.52± 0.77 | 48.74 ± 0.46 | 52.05 ± 0.16 |
| | LAGCN | 64.09 ± 0.65 | 69.62 ± 0.25 | 72.08 ± 0.14 | 47.12± 0.63 | 49.45 ± 0.43 | 52.63 ± 0.16 |
| | NodeMixup | 63.91 ± 0.87 | 69.85 ± 0.23 | 73.26 ± 0.25 | 46.65± 1.66 | 48.92 ± 0.56 | 52.54 ± 0.21 |
| | IntraMix | **64.84 ± 0.38** | **70.21 ± 0.17** | **73.51 ± 0.22** | **48.18± 0.68** | **50.13 ± 0.32** | **53.03 ± 0.25** |
| GAT | Original | 63.21 ± 0.94 | 69.75 ± 0.43 | 73.65 ± 0.11 | 45.88 ± 1.23 | 48.24 ± 0.53 | 49.88 ± 0.32 |
| | FLAG | 63.80 ± 0.84 | 69.93 ± 0.51 | 73.71 ± 0.13 | 46.24 ± 0.75 | 48.51 ± 0.43 | 51.34 ± 0.27 |
| | LAGAT | 64.21 ± 0.54 | 69.96 ± 0.22 | 73.77 ± 0.12 | 47.55 ± 0.65 | 49.65 ± 0.28 | 52.63 ± 0.16 |
| | NodeMixup | 64.26 ± 0.47 | 70.05 ± 0.21 | 73.24 ± 0.32 | 47.02 ± 1.29 | 49.05 ± 0.61 | 52.82 ± 0.36 |
| | IntraMix | **65.01 ± 0.35** | **70.73 ± 0.20** | **73.85 ± 0.12** | **48.32 ± 0.61** | **50.33 ± 0.37** | **53.49 ± 0.09** |
| SAGE | Original | 62.87± 0.81 | 68.82 ± 0.40 | 71.49 ± 0.27 | 45.72 ± 1.12 | 48.65 ± 0.43 | 50.47 ± 0.21 |
| | FLAG | 63.35 ± 0.77 | 69.04 ± 0.38 | 72.19 ± 0.21 | 46.54 ± 0.78 | 48.23 ± 0.43 | 52.39 ± 0.28 |
| | LASAGE | 64.26 ± 0.57 | 69.93 ± 0.24 | 72.30 ± 0.12 | 47.33 ± 0.63 | 50.82 ± 0.38 | 54.24 ± 0.25 |
| | NodeMixup | 64.01 ± 0.44 | 69.79 ± 0.23 | 72.01 ± 0.35 | 47.13 ± 0.58 | 50.91 ± 0.24 | 53.49 ± 0.24 |
| | IntraMix | **65.32 ± 0.26** | **70.56 ± 0.22** | **73.61 ± 0.09** | **48.24 ± 0.51** | **51.42 ± 0.29** | **54.65 ± 0.26** |

Additionally, Table 2 shows that the semi-supervised results with 1% and 5% of the training data on large datasets also demonstrates excellent performance. This indicates that the IntraMix generation of high-quality labeled nodes and neighborhoods, enriches the knowledge on the graph, making the graph more conducive for GNNs. Moreover, it is noteworthy that IntraMix exhibits greater advantages on SAGE and APPNP. This is attributed to the neighbor sampling for message aggregation of SAGE and the customized message-passing of APPNP, both of which prioritize the correct and richness of neighborhoods. The superiority on these two models further validates the rationality and richness of the neighborhoods constructed by IntraMix and the correctness of the generated nodes.

## 4.2 Full-supervised Learning

**Datasets:** To evaluate IntraMix on full-supervised datasets, we utilize the well-known ogbn-arxiv and Flickr, following standard partitioning ways. Detailed information can be found in Appendix C.1.

**Baselines:** In this part, we consider three GNNs: GCN, GAT, and GraphSAGE. We compare IntraMix with various mainstream methods, and details about these methods can be found in C.2.

**Results:** The results in Table 2 show that IntraMix consistently outperforms all GNNs and datasets in full-supervised experiments, aligning with the outcomes in semi-supervised learning. Although graphs typically adhere to semi-supervised settings, some graphs, like citation networks, have sufficient labels [34]. Thus, we conduct supervised experiments to show that the generality of IntraMix. The success in full-supervised settings primarily demonstrates the effectiveness of our neighbor selection strategy, as the ample labeled data in the training set reduces the influence of the high-quality labeled data generated by IntraMix. This further proves that our neighbor selection strategy constructs a graph more conducive to downstream tasks by enriching the high-quality neighborhoods of nodes.

## 4.3 Inductive Learning

The experiments mentioned above are conducted in transductive settings. In node-level tasks, the common setting is transductive, where the test distribution is known during training, fitting many static graphs. Inductive learning refers to not knowing the test distribution during training. Since many real-world graphs are dynamic, inductive learning is also crucial. To demonstrate the reliability of IntraMix in inductive setups, we conduct experiments on Cora and CiteSeer, utilizing GraphSAGE and GAT. The results are presented in Table 3. In inductive learning, GNNs can only observe non-test data during training.

Table 3: Node Classification in inductive settings.

| Models | Strategy | Cora | CiteSeer |
|---|---|---|---|
| GAT | Original | 81.3 ± 0.5 | 70.4 ± 1.2 |
| | LAGAT | 82.7 ± 0.8 | 72.1 ± 0.7 |
| | NodeMixup | 83.1 ± 0.5 | 71.8 ± 0.9 |
| | IntraMix | **83.8 ± 0.6** | **72.9 ± 0.6** |
| SAGE | Original | 80.1 ± 1.7 | 69.1 ± 2.9 |
| | LAGSAGE | 81.7 ± 0.8 | 73.0 ± 1.1 |
| | NodeMixup | 81.9 ± 0.5 | 73.1 ± 1.3 |
| | IntraMix | **82.9 ± 0.4** | **73.9 ± 0.8** |

Table 4: Ablation of Intra-Class Mixup on GCN. *w con* is vallina mixup connection, and *sim con* is similar connection. ↑ is the improvement.

| Strategy | Cora | CiteSeer | Pubmed |
|---|---|---|---|
| Original | $81.5 \pm 0.4$ | $70.3 \pm 0.5$ | $79.0 \pm 0.3$ |
| Only PL | $82.9 \pm 0.2$ | $72.3 \pm 0.3$ | $79.5 \pm 0.2$ |
| Only UPS | $83.1 \pm 0.4$ | $72.8 \pm 0.6$ | $79.7 \pm 0.4$ |
| Mixup(w/o con) | $58.9 \pm 22.3$ | $52.3 \pm 17.6$ | $70.0 \pm 10.8$ |
| Mixup(w con) | $83.0 \pm 1.2$ | $71.3 \pm 3.5$ | $79.4 \pm 1.1$ |
| Mixup(sim con) | $83.1 \pm 1.8$ | $71.5 \pm 1.9$ | $79.8 \pm 3.8$ |
| Intra-Class Mixup | **85.2 (↑3.7)** | **74.8 (↑4.5)** | **82.9 (↑3.9)** |

Table 5: Effects of Neighbor Selection on GCN. ↑ means improvement compared to the original, while ↓ indicates a reduction.

| Strategy | Cora | CiteSeer | Pubmed |
|---|---|---|---|
| Original-GCN | $81.5 \pm 0.4$ | $70.3 \pm 0.5$ | $79.0 \pm 0.3$ |
| Direct Con | 83.6 (↑ 2.1) | 73.4 (↑ 3.1) | 78.0 (↓ 1.0) |
| Random Con | 76.7 (↓ 4.8) | 67.0 (↓ 3.3) | 65.1 (↓ 13.9) |
| Without Con | 82.9 (↑ 1.4) | 72.8 (↑ 2.5) | 79.4 (↑ 0.4) |
| Vallina Con | 84.3 (↑ 2.8) | 73.6 (↑ 3.3) | 79.8 (↑ 0.8) |
| Similar Con | 84.5 (↑ 3.0) | 74.0 (↑ 3.7) | 80.3 (↑ 1.3) |
| IntraMix | **85.2 (↑ 3.7)** | **74.8 (↑ 4.5)** | **82.9 (↑ 3.9)** |

From the results, it is evident that IntraMix also exhibits excellent performance in inductive learning settings. This strongly validates that the generated nodes with more high-quality labels and rich neighborhoods constructed by IntraMix indeed provide the graph with more information beneficial to downstream tasks. As a result, GNNs trained with IntraMix can learn more comprehensive patterns and make accurate predictions even for unseen nodes, confirming IntraMix as a generalizable graph augmentation framework applicable to real-world scenarios.

## 4.4 Ablation Experiment

To demonstrate the effects of each IntraMix component, we conduct detailed ablation experiments using GCN on Cora, CiteSeer, and Pubmed. All other parts of IntraMix are kept unchanged except for the mentioned ablated components.

**Intra-Class Mixup:** Firstly, we use a simple experiment to show that part of the improvement is closely related to the high-quality data generated by Intra-Class Mixup. In Table 6, we replace the generated data with all-zero and all-one vectors and find that both perform worse than IntraMix. This indicates that the nodes generated by IntraMix are indeed helpful for downstream tasks.

Then, we discuss the effectiveness of Intra-Class Mixup. We compare it with methods that do not use Mixup, relying solely on pseudo-labeling(PL), and introduce an advanced PL method called UPS [32]. Additionally, we compare Intra-Class Mixup with vallina Mixup, which employs various connection methods. The results are shown in Table 4. Among these methods, Intra-Class Mixup has the best performance, showing nearly 3.5% improvement in accuracy compared to the original GCN. This is because, compared to methods using only pseudo-labels, Intra-Class Mixup generates higher-quality labeled nodes, enabling GNNs to get more information useful for downstream tasks.

Table 6: Explore the effect of generating node with Intra-Class Mixup. *Zeros* means replacing the generated nodes with an all-zero vector, and *Ones* means replacing them with an all-one vector.

| Strategy | Cora | CiteSeer | Pubmed |
|---|---|---|---|
| Original | $81.5 \pm 0.4$ | $70.3 \pm 0.5$ | $79.0 \pm 0.3$ |
| Ones | 31.9 (↓ 49) | 21.5 (↓ 48) | 38.1 (↓ 40) |
| Zeros | 83.8 (↑ 2.3) | 73.6 (↑ 3.3) | 80.7 (↑ 1.7) |
| IntraMix | **85.2 (↑3.7)** | **74.8 (↑4.5)** | **82.9 (↑3.9)** |

Regarding Mixup, we utilize three connecting methods: treating generated nodes as isolated (w/o con), connecting them with nodes used for generation (w con), and connecting them with nodes with similar embeddings (sim con). However, none of these methods perform well. As Theorem 3.1 suggests, Intra-Class Mixup ensures the improvement of label quality for each class, a guarantee that Mixup cannot provide. Furthermore, the fact that Intra-Class Mixup data have a single label makes it convenient to select similar neighbors. In contrast, Mixup generates data with mixed labels, introducing the risk of connecting to any class of node and potentially causing errors in propagation. This is a key reason for the poor performance of Mixup in node classification.

**Neighbor Selection:** This part shows the importance of Neighbor Selection. We compare various selection methods in Table 5. We observe that these methods are less effective than IntraMix. *Direct con* indicates connecting the generated data to low-quality labeled nodes of the same class, and its poor performance proves the necessity of our proposed approach to finding high-quality nodes of the same class as neighboring nodes. The experimental results validate Theorem 3.2.

Compared to other neighbor selection methods, IntraMix proposes a simple way to select nodes more likely to serve as neighbors, leading to more accurate message passing. Among the methods,

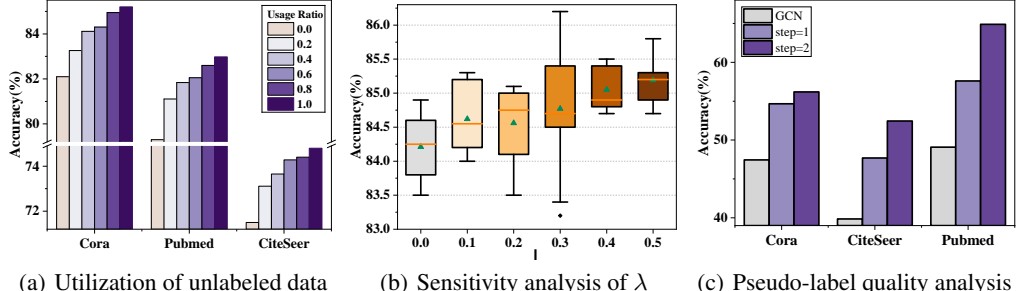

| (a) Utilization of unlabeled data | (b) Sensitivity analysis of $\lambda$ | (c) Pseudo-label quality analysis |

Figure 3: a) Experimental results using different proportions of unlabeled nodes show that performance improves as more unlabeled nodes are utilized. b) Sensitivity analysis of $\lambda$ indicates that the best performance is achieved when $\lambda = 0.5$. c) Analysis with low-quality pseudo-labels. The model from the previous step is used for pseudo-labeling in the next step.

*Vallina Con* indicates connecting generated nodes to the nodes used for generation. *Similar Con* (SC) denotes connecting the nodes to nodes with similar embeddings. SC performs great, highlighting the importance of selecting similar nodes as neighbors, aligning with our intuition. However, SC is not as good as IntraMix, mainly because the initial neighbors for generated nodes are empty, making it hard to provide accurate embeddings for similarity measurement. What's more, connecting overly similar nodes resulted in insufficient information. In comparison, IntraMix connects nodes with the same label, maintaining neighborhood correctness while connecting nodes that are not extremely similar. In Table 6, using an all-zero vector to eliminate the influence of Mixup still shows an improvement. This reflects the rationality of our neighbor selection method, which is effective for graphs.

**Utilization of unlabeled data:** In this part, we show the importance of using unlabeled nodes to obtain low-quality data, and the results are shown in Figure 3(a). Even though Mixup can augment the label information to some extent, the insufficient nodes used for generation create a bottleneck in information gain, hindering GNNs from learning enough knowledge. Despite the labels provided by pseudo-labeling for unlabeled data being low-quality, Intra-Class Mixup enhances the label quality, thus providing GNNs with ample knowledge.

**Sensitivity Analysis of $\lambda$:** This part discusses the impact of $\lambda$ in Intra-Class Mixup. The experiment is conducted using GCN on Cora, and details are presented in Figure 3 (b). According to Theorem 3.1, the best noise reduction in each class label is achieved when $\lambda = 0.5$. The results validate our theoretical analysis, showing that the performance of GCN gradually improves as $\lambda$ varies from 0 to 0.5. Therefore, we choose $\lambda \sim B(2, 2)$, where $B$ denotes Beta Distribution.

**Analysis of pseudo-label quality:** In this part we discuss the performance of IntraMix when the quality of pseudo-labels is extremely low. This situation may occur when the initial labeled nodes are extremely sparse. We use 5% of the semi-supervised training dataset for training. As shown in Figure 3 (c), we find that when the pseudo-label quality is low, IntraMix can effectively improve performance. Additionally, we use the model trained in the previous step for the next pseudo-labeling. This iterative method provides a way to enhance the performance of IntraMix on low-quality data. In summary, IntraMix effectively enriches the knowledge with extremely low-quality pseudo-labels.

## 4.5 Over-smoothing Analysis

As is well known, deep GNNs may result in over-smoothing, a phenomenon characterized by the convergence of node embeddings. We show the ability of IntraMix to alleviate over-smoothing in Figure 4(a). We use MADgap [3] as the metric, where a larger MADgap indicates a milder over-smoothing. Surprisingly, although IntraMix is not specifically designed to address over-smoothing, it shows a strong ability to counteract over-smoothing, reaching a level similar to GRAND [9], a method specialized in handling over-smoothing. This is attributed to the bridging effect of the generated nodes, connecting nodes of the same class with high confidence in a random manner. This process resembles random information propagation, providing effective resistance against over-smoothing. Additionally, the richer neighbors and node features generated by IntraMix inherently mitigate over-smoothing [17]. The detailed discussion can be found in Appendix D.1.

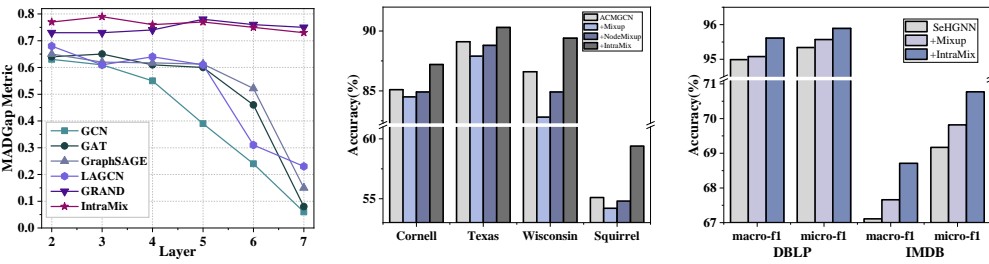

|(a) Over-smoothing Analysis|(b) Heterophilic Graphs Analysis|(c) Heterogeneous Graphs Analysis|

Figure 4: a) Analysis reveals that IntraMix shows effective capabilities in overcoming over-smoothing with deep GNNs. b) Evaluation on heterophilic graphs. c) Evaluation on heterogeneous graphs.

## 4.6 Evaluation on Heterophilic Graphs

In this section, we analyze the performance of IntraMix on heterophilic graphs on the Cornell, Texas and Wisconsin datasets [4]. Although the neighbor selection utilizes the neighborhood assumption, we find from the results in Figure 4(b) that IntraMix can also enhance GNN on heterophilic graphs. This is because, despite the existing connections in heterophilic graphs tending to link different types of nodes, they do not exclude connections between similar nodes. The connections between high-quality nodes generated by IntraMix can increase the information on the graph, thereby improving the performance of GNN. More detailed discussion is in Appendix D.2.

## 4.7 Evaluation on Heterogeneous Graphs

In this section, we discuss the performance of IntraMix on heterogeneous graphs. In this setting, neighboring nodes may belong to different types of entities. For example, different papers can be linked through authors, but there is no direct link between them. This is a common graph configuration. We use SeHGNN[54] as the base model and incorporate IntraMix to conduct experiments on the IMDB [47] and DBLP [38] datasets. The results are shown in Figure 4(c). IntraMix also improves performance in heterogeneous graphs, highlighting its versatility.

## 5 Related Work

**Graph Augmentation:** The primary purpose of graph augmentation is to address two common challenges in graphs encountered by GNN, scarcity of labels and incomplete neighborhoods [5]. Graph augmentation can be categorized into Node Manipulation [42], Edge Manipulation [33], Feature Manipulation [8], and Subgraph Manipulation [31]. However, existing methods either require complex generators [24] or extensive empirical involvement [44], failing to effectively address the two issues. The proposed IntraMix offers a simple solution to simultaneously tackle the two challenges faced by GNNs. Details about the related augmentation methods can be found in Appendix E.1.

**Mixup:** Mixup is a promising data augmentation medthod [50], enhancing the generalization of various tasks [41, 37]. However, there has been limited focus on the application in node classification on graphs. We address the shortcomings of vallina Mixup in node classification, proposing IntraMix. IntraMix provides richer information for graphs, improving GNNs in node classification. Details about the related Mixup works can be found in Appendix E.2.

## 6 Conclusion

This paper presents IntraMix, an elegant graph augmentation method for node classification. We utilize Intra-Class Mixup to generate high-quality labels to address the issue of sparse high-quality labels. To address the problem of limited neighborhoods, we connect the generated nodes with nodes that are highly likely from the same class. IntraMix provides an elegant solution to the dual challenges faced by graphs. Moreover, IntraMix is a flexible method that can be applied to all GNNs. Future work will focus on exploring neighbor selection methods to construct more realistic graphs.

## Acknowledgements

This work is supported by National Natural Science Foundation of China (NSFC) (62232005, 62202126); the National Key Research and Development Program of China (2021YFB3300502).

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

## A Notations

We first list the notations for key concepts in our paper.

Table 7: Notations

| Notations | Descriptions |
|---|---|
| G | A graph. |
| V | The set of nodes in a graph. |
| A | The graph adjacency matrix. |
| E | The set of edges in a graph. |
| $X \in R^{n \times d}$ | The feature matrix of nodes in the graph. |
| $Y \in R^{n \times c}$ | The labels of nodes in the graph. |
| $D_l$ | The set of nodes with labels at begining. |
| $D_u$ | The set of nodes without labels at begining. |
| $D_p$ | The set of nodes with low-quality labels. |
| $D_h$ | The set of nodes with high-quality labels. |
| $D_m$ | The set of nodes generated by Intra-Class Mixup. |
| $D_T$ | The set of nodes for training. |
| $N_v$ | The neighbors of a node $v$. |
| $h_v$ | The hidden feature vector of node $v$. |
| n | The number of nodes |
| m | The number of edges |
| $c$ | The number of node categories. |

## B Proofs

### B.1 Theorem 3.1

In this part, we will prove Theorem 3.1. We will start by introducing the notation used in the proof.

#### B.1.1 Notation and preliminaries

The category-specific noise is represented as:

$$P_{noise}(y_i|x) = P(y_i|x) + \epsilon_i \tag{6}$$

where $\epsilon_i \sim N(0, \sigma_i^2)$.

For simplification of the solution, we use the expression involving the label matrix transfer [28], denoted as $(x, y) \sim Q$. And get the noise label matrix transfer as:

$$Q_{noise} = T^t Q \tag{7}$$

where $T_{y,y'} \geq 0$ represents the probability, under the influence of noise, that data with the true label $y$ is mislabeled as $y'$.

And $T$ can be represented as :

$$T = (1 - f(\epsilon))I + f(\epsilon)J \tag{8}$$

Here, $\epsilon = \epsilon_i \sim N(0, \sigma_i^2)_{i=1}^{|C|}$ represents the noise for each class, $I$ denotes the diagonal matrix, and $J$ represents the matrix of all ones. Therefore, $f(\epsilon)J$ is the noise matrix, and the purpose of $f$ is to control the range of noise. It satisfies three properties: 1).$0 \leq f(x) \leq 1$; 2).$f(x)$ exhibits the same monotonicity as $|x|$; 3).$f(x) + f(y) = f(x + y)$;

Satisfying property 1 is essential because, in Equation 8, $f$ is used to control the range of noise, and probabilities represented by matrix entries should remain within the meaningful range of [0, 1].

Satisfying property 2 is due to the representation in Equation 6, where $|\epsilon_i|$ represents the magnitude of the deviation from the correct distribution, indicating larger noise for greater magnitudes. The positive or negative sign of $\epsilon_i$ indicates the direction of the noise, but in Equation 8, this directionality is not explicitly represented. Therefore, using the absolute value captures the magnitude of noise.

Satisfying property 3 is because the noise in Equation 6 has directionality; two noises may cancel each other out due to opposite directions. When represented using Equation 8, considering the directional of noise, operations need to be performed first, which might lead to cancellation due to different directions. Afterward, applying $f$ is necessary to ensure the result falls within the (0, 1) range, adhering to the definition of probabilities. For simplicity, we assume that $f$ directly satisfies property 3.

### B.1.2   Details of proof

**When we use Intra-Class Mixup**, According to Equation 6, the distribution of labels for the generated data by Intra-class Mixup can be expressed as:

$$P_{intra}(y_i|\hat{x}) = \lambda(P(y_i|\hat{x}) + \epsilon_i) + (1 - \lambda)(P(y_i|\hat{x}) + \epsilon_i') \tag{9}$$

where $\epsilon_i \sim N(0, \sigma_i^2)$, $\epsilon_i' \sim N(0, \sigma_i^2)$, and $\hat{x}$ represents the generated sample.

According to the definition of Intra-class Mixup and equation 8, the label transition matrix for the results of Intra-class Mixup can be expressed as:

$$T_{intra} = (1 - f(\lambda\epsilon_1 + (1 - \lambda)\epsilon_2))I + f(\lambda\epsilon_1 + (1 - \lambda)\epsilon_2)J \tag{10}$$

where $\epsilon_1 = \{\epsilon_i \sim N(0, \sigma_i^2)\}_{i=1}^{|C|}$ and $\epsilon_2 = \{\epsilon_i \sim N(0, \sigma_i^2)\}_{i=1}^{|C|}$.

The measurement of noise in $T$ and $T_{intra}$ is essentially comparing $f(\epsilon)$ with $f(\lambda\epsilon_1 + (1 - \lambda)\epsilon_2)$. According to property 2 satisfied by $f$, this can be transformed into comparing $|\epsilon|$ with $|\lambda\epsilon_1 + (1 - \lambda)\epsilon_2|$. A larger value in this comparison indicates greater noise. For the $i$-th class label, it is a calculation of:

$$P(|\lambda\epsilon_{i1} + (1 - \lambda)\epsilon_{i2}| < |\epsilon_i|) \tag{11}$$

where $\epsilon_i \sim N(0, \sigma_i^2)$, $\epsilon_i, \epsilon_{i1}, \epsilon_{i2}$ are independent and identically distributed variables, and $|\epsilon|$ follows the folded normal distribution [22], let $c = \lambda^2 + (1 - \lambda)^2$, we have:

$$
\begin{aligned}
P(|\lambda\epsilon_{i1} + (1 - \lambda)\epsilon_{i2}| < |\epsilon_i|) &= \int_0^{+\infty} \frac{\sqrt{2}}{\sqrt{\pi}\sigma_i} e^{-\frac{x^2}{2\sigma^2}} \left( \int_0^{+\infty} \frac{\sqrt{2}}{\sqrt{\pi}c\sigma_i} e^{-\frac{y^2}{2c^2\sigma^2}} dy \right) dx \\
&\xlongequal[v=\frac{y}{\sigma_i}]{u=\frac{x}{\sigma_i}} \frac{2}{\pi c} \int_0^{+\infty} \int_0^u e^{-\frac{u^2}{2} - \frac{v^2}{2c^2}} dv du \\
&\xlongequal[v=r\sin\theta]{u=r\cos\theta} \frac{2}{\pi c} \int_0^{\frac{\pi}{4}} \int_0^{+\infty} re^{-\frac{r^2}{2} - \frac{1-c^2}{2c^2}r^2\sin^2\theta} dr d\theta \\
&= \frac{2}{\pi c} \int_0^{\frac{\pi}{4}} \frac{c^2}{c^2 + (1 - c^2)\sin^2\theta} d\theta \\
&\xlongequal{t=\tan\theta} \frac{2c}{\pi} \int_0^1 \frac{1}{c^2 + t^2} dt \\
&= \frac{2}{\pi} arctan\frac{1}{c} \\
&= \frac{2}{\pi} arctan(\lambda^2 + (1 - \lambda)^2)^{-\frac{1}{2}} > 0.5
\end{aligned}
\tag{12}
$$

Proof completed. This establishes that the likelihood of improving label quality after Intra-Class Mixup for each class is greater than 0.5, demonstrating its effectiveness in enhancing label quality.

Next, let's consider the expectation of noise, which can be equivalently expressed in terms of the expectation of $|\epsilon|$. Then we have:

$$
\begin{aligned}
E(|\lambda\epsilon_{i1} + (1 - \lambda)\epsilon_{i2}|) &= \int_0^{+\infty} x \frac{\sqrt{2}}{\sqrt{\pi}c\sigma_i} e^{-\frac{x^2}{2c^2\sigma_i^2}} dx \\
&\xlongequal{u=\frac{x}{c\sigma_i}} \frac{\sqrt{2}c\sigma_i}{\sqrt{\pi}} \int_0^{+\infty} ue^{-\frac{u^2}{2}} du \\
&= \frac{\sqrt{2}\sigma_i}{\sqrt{\pi}}
\end{aligned}
\tag{13}
$$

Similarly, we have: $E(|\epsilon_i|) = \frac{\sqrt{2}\sigma}{\sqrt{\pi}}$.

Therefore, the ratio of the expected noise after Intra-Class Mixup to the original noise in class $i$ is given by:

$$\frac{E(f(\lambda\epsilon_{i1} + (1-\lambda)\epsilon_{i2}))}{E(f(\epsilon_i))} \sim \frac{E(|\lambda\epsilon_{i1} + (1-\lambda)\epsilon_{i2}|)}{E(|\epsilon_i|)} = \sqrt{\lambda^2 + (1-\lambda)^2} < 1 \tag{14}$$

This implies that the expected noise in each class is reduced after Intra-Class Mixup.

**When we use Vallina Mixup**, i.e., performing Mixup randomly between samples, it can be expressed according to the derivation in [2] and Equation 8 as follows:

$$T_{mixup} = (1 - f(\lambda\epsilon_1 + (1-\lambda)\bar{\epsilon}))I + f(\lambda\epsilon_1 + (1-\lambda)\bar{\epsilon})J \tag{15}$$

where $\bar{\epsilon} = \frac{1}{|C|}\sum_{i=1}^{|C|}\epsilon_i$.

Then, following similar derivations of Equation 12, let $\sigma_{i2} = \sqrt{\lambda^2\sigma_i^2 + (1-\lambda)^2\bar{\sigma}^2}$, $\bar{\sigma} = \frac{1}{|C|}\sum_{i=1}^{|C|}\sigma_i$, we have:

$$\begin{aligned}
P(|\lambda\epsilon_1 + (1+\lambda)\bar{\epsilon}| < |\epsilon|) &= \int_0^{+\infty} \frac{\sqrt{2}}{\sqrt{\pi}\sigma_i}e^{-\frac{x^2}{2\sigma^2}}\left(\int_0^{+\infty}\frac{\sqrt{2}}{\sqrt{\pi}c\sigma_i}e^{-\frac{y^2}{2c^2\sigma_{i2}^2}}\,dy\right)dx \\
&= \frac{2}{\pi}arctan\frac{\sigma_i}{\sqrt{\lambda^2\sigma_i^2 + (1-\lambda)^2\bar{\sigma}^2}}
\end{aligned} \tag{16}$$

The Equation 16 cannot guarantee that it is greater than 0.5 for all $\sigma_i$. This implies that although Vallina Mixup acts as a form of regularization during training, implicitly introducing data denoising [50], it does not ensure label quality improvement for every class. In this regard, it has limitations compared to our proposed Intra-Class Mixup.

## B.2 Theorem 3.2

In this section, we will prove Theorem 3.2. We will begin by introducing the notation used in the proof process.

### B.2.1 Notation and preliminaries

In the proof of this section, the labeling noise in Equation 6 is transformed into equivalent node feature noise as follows:

$$P_{noise}(x|y_i) = P(x|y_i) + \delta_i \tag{17}$$

where $\delta_i \sim N(0, \sigma_{xi}^2)$. The equivalence proof with Equation 6 is as follows:

$$\begin{aligned}
P_{noise}(x|y_i) &= P_{noise}(y_i|x)\frac{P(x)}{P(y_i)} \\
P(x|y_i) &= P(y_i|x)\frac{P(x)}{P(y_i)}
\end{aligned} \tag{18}$$

Therefore, from Equation 17, we have:

$$\begin{aligned}
P_{noise}(x|y_i) &= P_{noise}(y_i|x)\frac{P(x)}{P(y_i)} \\
&= (P(y_i|x) + \epsilon_i)\frac{P(x)}{P(y_i)} \\
&= (P(x|y_i)\frac{P(y_i)}{P(x)} + \epsilon_i)\frac{P(x)}{P(y_i)} \\
&= P(x|y_i) + \frac{P(x)}{P(y_i)}\epsilon_i
\end{aligned} \tag{19}$$

$\frac{P(x)}{P(y_i)}$ is a constant related to the dataset. Thus, the equation $P_{noise}(x|y_i) = P(x|y_i) + \delta_i$ holds, where $\delta_i \sim N(0, \sigma_{xi}^2)$. The equivalence between label noise and feature noise holds.

In this section, for the sake of convenience in derivation, the information propagation formula of the GNN in Equation 1 at the $k$-th layer is simplified to:

$$h_v^k = MLP^k[(1 + \eta_k)h_v^{k-1} + \frac{1}{|N_v|} \sum_{u \in N_v} h_u^{k-1}] \tag{20}$$

Where $h_v^k$ is used to represent the feature representation of node $v$ at the $k$-th layer, $\eta_k$ represents the learnable variable at the $k$-th layer, $N_v$ denotes the set of neighboring nodes of node $v$. $W_k$ can be used to represent the parameter matrix of the MLP in the $k$-th layer of the GNN, and $b_k$ represents the bias term of the MLP in the $k$-th layer.

In this part, consider two nodes $m$ and $n$, both belonging to class $y_i$ under high-confidence labeling. Node $m$ has a node feature $x_m$ following the distribution $P(x|y_i)$, while the node feature $x_n$ of node $n$ follows the distribution $P_{noise}(x|y_i)$. We have:

$$P(x_m|y_i) = P(x|y_i), P(x_n|y_i) = P_{noise}(x|y_i) = P(x|y_i) + \delta_i \tag{21}$$

For convenience in derivation, without loss of generality, we can assume that $m$ and $n$ have no neighbors in their initial states. We will prove that in a two-layer GNN: 1). By the approach proposed through IntraMix, connecting $m$ and $n$ through the node $v = (\hat{x}, y_i)$ generated by Intra-Class Mixup. 2). Connecting nodes $m$ and $n$ directly. The ratio of the expected impact of $n$'s noise on $m$ in case 1) to the expected noise impact in case 2) can be controlled to be less than 1 by adjusting learnable $\eta$. Therefore, the connection approach in IntraMix can to some extent overcome the potential noise disturbance that may exist when connecting high-quality labeled nodes directly.

### B.2.2 Details of proof

**When connecting nodes $m$ and $n$ directly, we have:**

$$\begin{aligned}
P(h_m^1|y_i) &= W_1[(1 + \eta_1)P(x_m|y_i) + P(x_n|y_i)] + b_1 \\
&= W_1[(2 + \eta_1)P(x|y_i) + \delta_i] + b_1
\end{aligned} \tag{22}$$

Similarly, we have:

$$P(h_n^1|y_i) = W_1[(2 + \eta_1)P(x|y_i) + (1 + \eta_1)\delta_i] + b_1 \tag{23}$$

Then, after the second-layer message passing, node $m$ can be represented as:

$$\begin{aligned}
P(h_m^2|y_i) &= W_1[(1 + \eta_2)P(h_m^1|y_i) + P(h_n^1|y_i)] \\
&= W_2\{(1 + \eta_2)[W_1[(2 + \eta_1)P(x|y_i) + \delta_i] + b_1] \\
&\quad + W_1[(2 + \eta_1)P(x|y_i) + (1 + \eta_1)\delta_i] + b_1\} + b_2 \\
&= \underbrace{W_2\{(2 + \eta_1)[(1 + \eta_2)W_2W_1 + W_1]P(x|y_i) + (2 + \eta_2)b_1\} + b_2}_{value} \\
&\quad + \underbrace{W_2W_1(2 + \eta_1 + \eta_2)\delta_i}_{noise}
\end{aligned} \tag{24}$$

**When connecting through Intra-Class Mixup with the node $v$, we have:**

According to Equation 17 and Equation 10, the feature distribution of the node $v$ generated by Intra-Class Mixup satisfies $P(\hat{x}|y_i) = P(x|y_i) + \delta_i'$, where $\delta_i' \sim N(0, (\lambda^2 + (1-\lambda)^2)\sigma_{xi}^2)$.

Following the same reasoning as above, we have:

$$\begin{aligned}
P'(h_m^1|y_i) &= W_1[(1 + \eta_1)P(x_m|y_i) + P(\hat{x}|y_i)] + b_1 \\
&= W_1[(2 + \eta_1)P(x|y_i) + \delta_i'] + b_1 \\
P'(h_n^1|y_i) &= W_1[(2 + \eta_1)P(x|y_i) + \delta_i' + (1 + \eta_1)\delta_i] + b_1 \\
P'(h_v^1|y_i) &= W_1[(1 + \eta_1)P(\hat{x}|y_i) + \frac{1}{2}P(h_m^0|y_i) + \frac{1}{2}P(h_n^0|y_i)] + b_1 \\
&= W_1[(2 + \eta_1)P(x|y_i) + \delta_i' + \frac{1}{2}\delta_i] + b_1
\end{aligned} \tag{25}$$

Then, after the second-layer message passing, $m$ can be represented as:

$$
\begin{aligned}
P'(h_m^2|y_i) &= W_1[(1+\eta_2)P'(h_m^1|y_i) + P'(h_v^1|y_i)] \\
&= W_2\{(1+\eta_2)[W_1[(2+\eta_1)P(x|y_i) + \delta_i'] + b_1] \\
&\quad + W_1[(2+\eta_1)P(x|y_i) + \delta_i' + \frac{1}{2}\delta_i] + b_1\} + b_2 \\
&= \underbrace{W_2\{(2+\eta_1)[(1+\eta_2)W_2W_1 + W_1]P(x|y_i) + (2+\eta_2)b_1\} + b_2}_{value} \\
&\quad + \underbrace{W_2W_1[(2+\eta_1+\eta_2)\delta_i' + \frac{1}{2}\delta_i]}_{noise}
\end{aligned}
\tag{26}
$$

Assuming the parameters of MLP are the same, the value parts of Equation 24 and Equation 26 are the same. Now, what needs to be compared is the ratio of the expected values of the noise parts in the two equations. Similar to property 3 mentioned in Sec B.1.1, when comparing noise expectations, absolute values should be considered. That is, the farther the noise is from the 0, the larger it is, and the sign indicates the direction. Let $noise_m$ and $noise_m'$ represent the noise terms in Equation 24 and Equation 26, respectively. We have:

$$
\frac{E(noise_m')}{E(noise_m)} = \frac{E\{|(2+\eta_1+\eta_2)\delta_i' + \frac{1}{2}\delta_i|\}}{E\{|(2+\eta_1+\eta_2)\delta_i|\}}
\tag{27}
$$

Similar to the derivation in Equation 13, we have:

$$
\begin{aligned}
E\{|(2+\eta_1+\eta_2)\delta_i|\} &= \frac{\sqrt{2}(2+\eta_1+\eta_2)\sigma_{xi}}{\sqrt{\pi}} \\
E\{|(2+\eta_1+\eta_2)\delta_i' + \frac{1}{2}\delta_i|\} &= \frac{\sqrt{2}\sigma_{xi}\sqrt{\frac{1}{4} + (2+\eta_1+\eta_2)^2(\lambda^2+(1-\lambda)^2)}}{\sqrt{\pi}}
\end{aligned}
\tag{28}
$$

Then, Equation 27 can be represented as:

$$
\frac{E(noise_m')}{E(noise_m)} = \sqrt{\frac{1}{4(2+\eta_1+\eta_2)} + (\lambda^2+(1-\lambda)^2)}
\tag{29}
$$

As $\eta_1$ and $\eta_2$ are learnable parameters, by controlling them, the value of Equation 29 can be made less than 1. This indicates that connecting nodes obtained through Intra-Class Mixup leads to smaller noise compared to directly connecting them.

## C Reproducibility

In this section, we present detailed information about the datasets used in the experiments, including the split method. Additionally, we provide a thorough introduction to the comparative methods and the setups of GNNs in the experiments.

### C.1 Datasets

In this part, we introduce the datasets used in this paper. Detailed information can be found in Table 8.

We utilize five medium-scale datasets in a semi-supervised setting: Cora, CiteSeer, Pubmed [34], CS, and Physics [35]. All five datasets are related to citation networks. The first three datasets are standard citation datasets, where nodes represent papers, edges between nodes indicate citation relationships between papers, and node features are constructed by extracting key information from the papers, such as a one-hot vector indicating the presence of specific words in the paper. Node classification on these three datasets involves assigning papers to their respective categories. During the dataset split process, we follow the standard partitioning method outlined in the literature [34], using a minimal amount of samples for training to adhere to the semi-supervised configuration.

On the other hand, CS and Physics datasets represent datasets related to collaboration among researchers. In these datasets, each node represents a researcher, and edges denote collaborative

Table 8: Data statistics

| Category | Name | Graphs | Nodes | Edges | Features | Classes | Split Ratio | Metric |
|---|---|---|---|---|---|---|---|---|
| | Cora | 1 | 2,708 | 5,429 | 1,433 | 7 | 8.5/30.5/61 | Accuracy |
| | CiteSeer | 1 | 3,327 | 4,732 | 3,703 | 6 | 7.4/30.9/61.7 | Accuracy |
| Semi-Supervised | Pubmed | 1 | 19,717 | 44,338 | 500 | 3 | 3.8/32.1/64.1 | Accuracy |
| | CS | 1 | 18,333 | 163,788 | 6,805 | 15 | 1.6/2.4/96 | Accuracy |
| | Physics | 1 | 34,493 | 495,924 | 8,415 | 5 | 0.28/0.43/99.29 | Accuracy |
| Full-Supervised | ogbn-arxiv | 1 | 169,343 | 1,166,243 | 128 | 40 | 54/18/28 | Accuracy |
| | Flickr | 1 | 89,250 | 899,756 | 500 | 7 | 50/25/25 | Accuracy |

relationships between researchers in co-authored papers. Node features capture partial characteristics of researchers' papers. The process of node classification involves assigning researchers to their respective research directions. During the dataset split process, we follow the standard partitioning method [35], randomly selecting 20 samples from each class as training samples and using 1000 samples in total for the training set to meet the semi-supervised configuration.

We also use two large-scale datasets for semi-supervised and full-supervised training: ogbn-arxiv [16] and Flickr [49]. **Their traditional splits are suitable for full-supervised training, while in the semi-supervised configuration, we used 5% and 10% of the original training data to simulate the semi-supervised setting.** The ogbn-arxiv is a dataset of the ogb standard dataset, where each node represents an Arxiv paper, and directed edges indicate citations between papers. Each paper is associated with a 128-dimensional feature vector obtained by averaging embeddings of words in the title and abstract. The classification task involves predicting the primary category of Arxiv papers, constituting a 40-class classification problem. The dataset partitioning follows the original paper [16], with papers published until 2017 serving as the training set, papers published in 2018 as the validation set, and papers published after 2019 as the test set. Unlike the limited training set in semi-supervised learning, the training set here is relatively larger.

On the other hand, Flickr is constructed by forming links between shared public images on Flickr. Each node in the graph represents an image uploaded to Flickr. If two images share certain attributes (e.g., the same geographic location, the same gallery, comments posted by the same user), there is an edge between the nodes representing these two images. The dataset is collected from various sources, and images are represented using SIFT-extracted features. A 500-dimensional bag-of-visual-words representation from NUS-wide serves as the node feature. For labels, the paper [49] scans 81 tags for each image and manually merges them into 7 categories. Each image belongs to one of the 7 categories. The dataset partitioning follows the approach proposed in the paper [49], with 50% of the nodes as the training set, 25% as the validation set, and 25% as the test set.

## C.2 Baselines

This section introduces the methods compared in the experiments. For the fairness of the experiments, we maintain consistent GNN structures when testing different augmentation methods. The configurations of GNNs are kept consistent with those described in [24]. We introduce several data augmentation methods compared in this paper:

**GraphMix** [42]: This is an effective exploration of using the Mixup method in graphs, attempting to perform Mixup in the hidden layer outputs of GNN to mitigate the impact of the topological structure of graphs on the Mixup. However, it overlooks the core issue we analyzed regarding Mixup on graphs: the inability to determine neighborhood relationships. The method used in this paper connects the generated sample with the samples used for generation, resulting in poor performance and only addressing the issue of missing high-quality nodes.

**CODA** [6]: The paper proposes to extract all nodes of the same class to generate a dense graph for each class, using dense graphs of each class as generated graphs. However, this approach simplifies the topological relationships on the graph, as not every node of the same class can appear as neighbors. Connecting each node of the same class makes the neighborhood very rich, but it leads to excessive message transfer that should not occur, resulting in poor performance.

**FLAG** [20]: This is a data augmentation method that uses advertial learnable noise perturbations, quite effective on large graphs. It incorporates learnable noise into node features and optimizes GNN training by finding the most challenging noise in each iteration. However, inevitably, it requires

fine-tuning between the magnitude of noise and its effectiveness in training, which can ultimately lead to noise being either too small or too large, both hindering learning. Moreover, it only addresses the issue of missing high-quality nodes.

**DropMessage** [7]: In this method, random dropout of information during message passing is considered, representing a unified form for drop node/drop edge-like strategies. However, as analyzed in the paper, this strategy is overly simplistic and fails to adapt well to graph augmentation tasks that require nuanced operations, resulting in poor performance.

**MH-Aug** [30]: This approach formulates a target distribution based on the existing graph, samples a series of augmented graphs from the distribution using a sampling method proposed in the paper, and trains using a consistency approach. However, this method introduces certain prior knowledge, which may lead to suboptimal generalization.

**GRAND** [9]: Strictly speaking, this is not a data augmentation method, but it shares some similarities in terms of ideas. Grand randomly propagates information on the graph, significantly mitigating issues like over-smoothing. While it presents a novel solution, the strategy used is overly simplistic and requires further optimization based on dataset characteristics to achieve better performance. Additionally, it only addresses the problem of neighborhood deficiency.

**Local Augmentation** [24]: This method employs the conditional variational autoencoder (CVAE) as the generator to learn information about neighborhoods and generates features for nodes with sparse neighbors to compensate for the lack of information that cannot be obtained from the neighborhood, thereby obtaining better node representations for classification. This approach incurs a training cost for the generator and only addresses the issue of missing high-quality nodes.

**NodeMixup** [25]: This is an exploratory approach to Mixup on graphs. However, it faces the challenge of not proposing an elegant and concise graph Mixup method. It focuses more on accurately determining node similarity to perform Mixup on similar nodes. The issue arises from the fact that the generated samples are excessively similar to each other, resulting in generated samples being relatively similar to their parent samples. Generated samples that are overly similar to the original samples fail to provide sufficient information gain. We will provide a detailed explanation in Sec E.1. Additionally, NodeMixup only addresses the issue of missing high-quality nodes.

In summary, existing methods often address only one aspect of the two challenges faced by graph data, failing to effectively solve the graph data augmentation problem. In contrast, our proposed method, IntraMix, presents a concise yet powerful solution that efficiently tackles both challenges, making it a highly promising approach.

## D    More Discussions and Future Directions

In this section, we explain some analyses in the main text and introduce future research directions.

### D.1    Over-smoothing Analysis

We further analyze the ability of IntraMix to resist over-smoothing. Over-smoothing results in the embeddings of different nodes on the graph become too similar, making it difficult for GNNs to distinguish between node classes based on the output layer. As the depth of the GNN increases, over-smoothing tends to worsen because deeper GNNs use a broader range of neighbors for each node's embedding process, leading to excessive overlap in the neighborhoods and resulting in very similar node embeddings [3]. However, an interesting phenomenon is that within a certain range, increasing the depth of the GNN does not reduce MADGap. This is because deeper GNNs have enhanced representational power, which can produce more distinct embeddings for each node despite the increased depth [29].

IntraMix is particularly impressive in its ability to resist over-smoothing to some extent. We attribute this phenomenon to two main reasons. First, IntraMix generates a large number of accurately labeled nodes, providing each node with numerous accurate neighbors. For any given node, a richer neighborhood naturally helps counteract over-smoothing [17]. Second, it is important to note that we use a random connection strategy during neighborhood selection. This means that nodes which might be far apart in the original graph can become connected through the generated nodes, bridging two neighborhoods that have some commonalities but are not very similar. This enriches the neighborhood

information and reduces the impact of over-smoothing. The second reason is somewhat similar to why GRAND [9] performs well in resisting over-smoothing.

## D.2 Evaluation on heterophilic graphs

In this part, we analyze the effectiveness of IntraMix in heterophilic graphs. Despite the assumption we use when finding neighbors, which is that nodes of the same class are more likely to appear in the neighborhood, contradicting the nature of heterophilic graphs, the results from Figure 4(b) show that IntraMix still enhances GNN performance in heterophilic graphs. This is because, although the inherent connectivity in heterophilic graphs tends to favor connections between nodes of the different classes, it does not exclude connections between nodes of the same class. Therefore, connections between nodes of the same class can also enrich the information of the graph. In other words, although GNNs learned on heterophilic graphs cannot assume that nodes of the same class are more likely to appear in the neighborhood, adding connections between nodes of the same class to such graphs does not undermine their inherent properties.

This can be easily understood using the WebKB dataset [4] as an example. This is a classic heterophilic graph dataset, with each node representing one of Student, Faculty, Staff, Department, Course, or Project. Without loss of generality, we only consider Student, Staff and Course. We can construct a simplified relationship as shown in the Figure. Here, the connection between Staff A and Student B can represent that A is the advisor of B, while the connection between Student B and Course C can represent that B is enrolled in Course C. The inherent connections tend to favor such connections between different classes. However, IntraMix generates connections such as the one between two students, $B_1$ and $B_2$, which also holds valid in reality and can represent a relationship between these two students. Intuitively, this relationship is logical and effective for downstream tasks. Through such relationships, IntraMix can enhance the effectiveness of the graph for downstream tasks. Therefore, IntraMix also has a certain effect on heterophilic graphs.

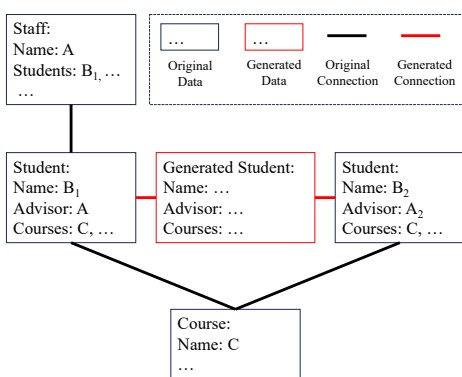

Figure 5: A simple example of IntraMix on WebKB. The original connections tend to link nodes of different classes, but the generated connections between nodes of the same class are also logically consistent.

## D.3 Analysis on Finding High-quality Nodes

In this part, we provide a further introduction to the method for finding high-quality labeled nodes mentioned in the main text and conduct a parameter sensitivity experiment. In the original text, we mentioned that we use multiple different dropout rates in a single GNN to approximate multiple distinct GNNs. By identifying nodes that are consistently labeled the same across all these GNN variants, we assume that these labels are approximately correct. This method serves as an alternative to ensemble learning. By using this approach, we avoid the need to pre-train multiple GNNs, and only incur the cost of inference n times, significantly reducing the time consumption.

We conduct a parameter sensitivity experiment on $n$, analyzing which value of n yields the best performance. The results are shown in the table. It can be observed that $n$ is not always better when larger; the optimal value is achieved at $n = 5$, while the performance declines at $n = 6$. This is quite understandable. As $n$ increases, it can be seen as an increase in the number of models used in ensemble learning, leading to more accurate node labeling. In other words, as $n$ increases, if the labels of a node are consistent across all $n$ inferences, the probability of it being correctly labeled increases. However, it is important to note that as $n$ increases, the number of nodes that are consistently labeled the same across all $n$

Table 9: Sensitivity Analysis of $n$. Experiments are conducted with GCN.

|     | Cora | CiteSeer | Pubmed |
| --- | --- | --- | --- |
| n=1 | 84.30 ± 0.60 | 73.60 ± 0.67 | 80.42 ± 0.23 |
| n=2 | 84.68 ± 0.74 | 73.86 ± 0.57 | 81.92 ± 0.43 |
| n=3 | 84.78 ± 0.54 | 74.01 ± 0.36 | 82.16 ± 0.28 |
| n=4 | 85.17 ± 0.55 | 74.41 ± 0.48 | 82.52 ± 0.38 |
| n=5 | **85.25 ± 0.42** | **74.80 ± 0.46** | **82.98 ± 0.54** |
| n=6 | 85.02 ± 0.65 | 74.15 ± 0.66 | 82.34 ± 0.60 |

Table 10: The comparison between IntraMix and Graph Structure Learning.

|  | Cora | CiteSeer | Pubmed |
|---|---|---|---|
| GCN | 81.51 ± 0.42 | 70.30 ± 0.54 | 79.06 ± 0.31 |
| SE-GSL [54] | 82.52 ± 0.51 | 71.08 ± 0.76 | 80.11 ± 0.45 |
| LAGCN | 84.61 ± 0.57 | 74.70 ± 0.51 | 81.73 ± 0.71 |
| IntraMix | 85.25 ± 0.42 | 74.80 ± 0.46 | 82.98 ± 0.54 |
| IntraMix+GAE | **85.99 ± 0.30** | **75.19 ± 0.33** | **83.52 ± 0.27** |

inferences decreases. This means that although the quality of these node labels is very high, their quantity is very low. Consequently, the feasible domain for selecting neighbors for the generated nodes becomes too small, failing to adequately enrich the information of graph. Therefore, in practical use, a trade-off needs to be made between these factors when choosing the best value of $n$.

### D.4 Comparison between IntraMix and Graph Structure Learning

In this section, we discuss the advantages of IntraMix compared to Graph Structure Learning(GSL). GSL is a method that optimizes graph structures and representations through learning. Therefore, it has overlap with data augmentation. Our biggest advantage over GSL methods lies in training and deployment costs. GSL methods require learnable ways for optimizing, leading to high training costs e.g., GAUG [51] uses GAE for edge probabilities, and Nodeformer [46] uses differentiable ways. In contrast, as analyzed in Sec 3.4, IntraMix has low time costs, making it practical for deployment. As real-world data continues to grow, the graphs in practical applications are becoming increasingly large, such as the expanding user networks in social media platforms. In this scenario, IntraMix offers a significant advantage in terms of training and deployment costs.

Secondly, in Table 10, we compare the performance of IntraMix and GSL. Additionally, we provided a method using IntraMix combined with GAE for structure selection, follow the setup of GAUG [51]. We find that IntraMix outperforms current popular GSL methods due to its effective use of low-quality labels and efficient topology optimization. At the same time, incorporating GAE improve the performance, but it also increase the training cost, presenting a trade-off.

In summary, IntraMix offers advantages over GSL methods with lower training and deployment costs and shows versatility by integrating various GSL methods.

### D.5 Future Directions

In this part, we discuss our plans for future work. As mentioned above, although IntraMix has shown some improvements in heterophilic graphs, it is evident that IntraMix was not specifically designed for heterophilic graphs. However, heterophilic graphs are quite common in real-world scenarios. Therefore, our future work will focus on better integrating IntraMix with heterophilic graphs. One possible approach is to gradually decrease the probability of connecting generated nodes with their surrounding neighbors during the training process. Initially, neighborhood selection satisfies homophily, meeting the requirements of this stage of homophilic graphs. Towards the end of this process, the generated nodes essentially exist as isolated nodes, abandoning the homophily assumption. This approach aims to satisfy the needs of both homophilic and heterophilic graphs. Further research is needed to explore this idea in detail.

## E Connection to Existing works

In this section, we present an expanded version of the Related Work Section 5.

### E.1 Graph Augmentation Methods

Graph data augmentation is a method aimed at addressing issues such as missing label data and incomplete neighborhoods in graph data. As mentioned in Sec 5, existing methods often suffer from various problems and are typically capable of addressing only one aspect of the problems, leaving the other unresolved [5]. This limitation is insufficient for the challenges posed by graph data augmentation. We believe that generating high-quality labeled data from low-quality samples holds potential as mentioned in Section 1. As noise in low-quality samples often causes the data

distribution to diverge in various directions [26], exceeding the intended distribution range, we plan to leverage this directionality of noise. By blending multiple samples through the direction of noise, we aim to neutralize the noise and generate high-quality samples. Thus, we propose IntraMix.

Although there are a few works that have recently focused on Mixup in the context of graph augmentation [42, 45, 25, 12, 43], they tend to overlook the unique characteristics of graphs, specifically the topological structure mentioned earlier. Most Mixup methods applied to graphs borrow strategies from image-based approaches, randomly mixing samples from different classes [45]. These approaches directly lead to the challenge of determining the neighborhood of generated samples. These works often choose to connect generated samples with the samples used for generation. This can result in problems due to the low-quality annotations on graphs, leading to incorrect message propagation. The subpar experimental results of these methods also substantiate this point [42].

While some works have attempted to use intra-class Mixup [25], they often treat it merely as a regularization component and do not realize its full potential. Similar to the use of inter-class Mixup, these approaches attempt to overcome the connectivity issues of Mixup by connecting nodes that have similar embeddings. However, this results in connected nodes being overly similar, providing limited meaningful information gain and leading to minimal performance improvement. These attempts highlight the inappropriate application of Mixup on graphs. Our proposed connection method ensures correctness in connections, while still providing rich information to a node's neighborhood since the similarity in labels does not necessarily imply high similarity in node features.

In summary, the proposed IntraMix elegantly addresses both the challenges of scarce high-quality labels and neighborhood incompleteness in graphs through a sophisticated Intra-Class Mixup approach and a high-confidence neighborhood selection method. The theoretical justification provided further supports the rationale behind this method.

## E.2 Mixup

Mixup is a data augmentation method that has demonstrated excellent performance in classical Euclidean data such as images [50], text [37], and audio [27]. The idea is to randomly sample data from the original dataset and mix the features and labels proportionally, generating new data. Both theoretically and experimentally, Mixup has been proven to enrich the distribution of the current data, allowing deep learning models to learn richer information [2, 1, 18]. Mixup can also be approximated as an implicit regularization during the training process [13].

However, in the case of images, each data exists independently, and there is no assumed relationship between individual samples. The independence and identical distribution (i.i.d.) assumption holds for image data. In contrast, graph data exhibits a neighbor relationship, and GNNs perform well due to this characteristic. This directly implies that when applying Mixup to graph data, the connectivity between generated samples and existing samples needs to be considered [45]. The full potential of GNNs can be realized only by constructing a reasonable neighborhood. This crucial aspect has been largely overlooked in almost all current Mixup approaches used in graphs.

Due to the necessity to consider connectivity and the assumption that neighboring nodes in the graph are more similar, a challenge arises when applying vallina Mixup: the generated data distribution often falls between the existing distributions of the two classes [36], making it difficult to resemble any specific class. Determining neighborhood relationships becomes challenging in this scenario. Hence, we innovatively propose conducting Mixup within the same class. Through this approach, the generated sample distribution substantially overlaps with the class used for generation, making it easier to determine neighborhoods. Existing Mixup methods typically discourage intra-class Mixup to preserve the diversity of generated samples. However, we break away from the empirical practice of existing Mixup techniques that only blend random samples from two different classes, presenting an elegant solution for graph data augmentation.

## F   Broader Impacts

This paper presents work whose goal is to advance the field of Graph Machine Learning. There are many potential societal consequences of our work, none which we feel must be specifically highlighted here.

