# OpenReview forum: "IntraMix: Intra-Class Mixup Generation for Accurate Labels and Neighbors"
_NeurIPS.cc/2024/Conference — NeurIPS 2024 poster_

### Official Review · Reviewer_bPiP · 2024-07-12

**Soundness:** 3
**Presentation:** 2
**Contribution:** 3
**Rating:** 5
**Confidence:** 4

**Summary:**

The paper introduces IntraMix, a novel method for augmenting graph data to improve the performance of Graph Neural Networks. IntraMix addresses two major issues in graph datasets: the lack of high-quality labeled data and inadequate neighborhood information. The proposed method leverages Intra-Class Mixup to generate high-quality labeled data from low-quality labels and uses a neighbor selection strategy to enrich the neighborhoods. Extensive experiments demonstrate that IntraMix enhances GNN performance across various datasets.

**Strengths:**

1. IntraMix effectively addresses both the lack of high-quality labels and inadequate neighborhood information, providing a comprehensive solution to two significant challenges in graph data augmentation.
2. The authors conduct thorough experiments on multiple datasets and GNN models, showing the robustness and effectiveness of the proposed method.
3. The paper provides theoretical analysis and guarantees for the reduction of noise in generated labels and the correctness of neighborhood enrichment, adding credibility to the method.
4. IntraMix is designed to be easily integrated into existing GNN frameworks without additional training costs, making it practical for real-world applications.

**Weaknesses:**

1. The method involves several steps, including pseudo-labeling, mixup, and neighbor selection, which might complicate the implementation and increase the computational overhead.
2. The initial pseudo-labeling step is crucial for the success of IntraMix. If the pseudo-labels are of very low quality, the subsequent steps may not generate significantly improved labels.
3. While the paper shows results on multiple datasets, the performance of IntraMix on graphs with highly heterogeneous or unusual structures is not thoroughly explored.
4. The method assumes a normal distribution for label noise, which may not hold in all real-world scenarios, potentially limiting the generalizability of the theoretical guarantees.

**Questions:**

See weaknesses.

**Limitations:**

Yes.

---

> ### Author Rebuttal · Authors · 2024-08-06
>
> Thank you for your constructive comments and suggestions. Next is our response.
>
> **Q1.The method involves several steps, which might complicate the implementation and increase the computational overhead.**
>
> A1. Concerns about complexity are reasonable but unnecessary for IntraMix.
>
> First, we analyze the implementation complexity. As detailed in Alg 1, only minimal pipeline modifications are needed.
>
> The time complexity analysis is in Sec 3.4. We break it down: **Pre-Training:** The model trained after data augmentation is the one for pseudo-labeling, so there is no great additional training cost. **Pseudo-labeling**: This only involves minimal inference overhead. **Mixup:** It only has a time complexity of O(m), where m is the number of generated data. **Neighbor Selection:** This requires only O(m) inferences, much less time-consuming than learnable ways. The time complexity for IntraMix is very low. Below, we present the time cost(s).
>
> |           | Pubmed    | ogbn-arxiv(5%) |
> | --------- | --------- | -------------- |
> | GCN       | 2.88±0.76 | 13.28±2.53     |
> | NodeMixup | 5.88±0.87 | 30.11±2.61     |
> | IntraMix  | 3.24±0.35 | 16.28±1.99     |
>
> The setting is consistent with Sec 4.1. Compared to GCN, IntraMix does not greatly increase costs, consistent with our analysis. We can reduce the generated data to further reduce costs.
>
> In summary, implementation and time complexity of IntraMix is not a concern.
>
> **Q2. If the pseudo-labels are of very low quality, the subsequent steps may not generate significantly improved labels.**
>
> A2. This is a great question. If the model cannot learn from data (e.g., pseudo-label accuracy is below 1%), IntraMix is likely to fail. However, such situations are extremely rare and instead of data augmentation, data collection is recommended in such cases. GNNs typically lead to reasonable pseudo-label quality.
>
> When pseudo-label quality is relatively low, IntraMix performs well. We use 5% of the semi-supervised training set for training.
>
> |          | Cora(5%)   | CiteSeer(5%) | Pubmed(5%) |
> | -------- | ---------- | ------------ | ---------- |
> | GCN      | 47.45±2.65 | 39.86±6.51   | 49.09±0.95 |
> | IntraMix | 54.66±1.61 | 47.70±2.13   | 57.60±0.83 |
>
> We find that when the pseudo-label quality is low, IntraMix can effectively improve performance.
>
> Additionally, iteration can improve performance. We use the model trained in the previous step for the next pseudo-labeling.
>
> |                           | Cora(5%)   | CiteSeer(5%) | Pubmed(5%) |
> | ------------------------- | ---------- | ------------ | ---------- |
> | step=0 (without IntraMix) | 47.45±2.65 | 39.86±6.51   | 49.09±0.95 |
> | step=1                    | 54.66±1.61 | 47.70±2.13   | 57.60±0.88 |
> | step=2                    | 56.19±1.54 | 52.45±2.89   | 64.88±0.82 |
>
> Iteration provides a way to enhance the performance of IntraMix on low-quality data.
>
> In summary, IntraMix effectively enriches the knowledge with low-quality pseudo-labels. Furthermore, iteration can help alleviate information bottlenecks for IntraMix.
>
> **Q3. The performance of IntraMix on graphs with highly heterogeneous or unusual structures.**
>
> A3. Thank you for this question. We conduct experiments to show that IntraMix is useful for heterogeneous graphs. In such data, nodes of the same type (e.g., movies in IMDB) often do not have direct connections. Therefore, we adjust our neighbor selection by connecting the neighbors of selected nodes. This minor modification does not impact the analysis.
>
> |           | DBLP       |            | IMDB       |            |
> | --------- | ---------- | ---------- | ---------- | ---------- |
> |           | macro-f1   | micro-f1   | macro-f1   | micro-f1   |
> | SeHGNN[1] | 94.99±0.19 | 95.34±0.14 | 67.11±0.25 | 69.17±0.43 |
> | +Mixup    | 95.08±0.31 | 95.57±0.21 | 67.66±0.31 | 69.82±0.45 |
> | +IntraMix | 95.61±0.22 | 95.89±0.10 | 68.71±0.26 | 70.77±0.32 |
>
> IntraMix also improves performance in heterogeneous graphs, highlighting its versatility. More details will be shown in the paper.
>
> **Q4. The method assumes a normal distribution for noise, limiting the generalizability.**
>
> This is a great question. We now analyze it.
>
> **Reasonableness:** Gaussian distribution is the most common noise distribution in reality, and many works[2,3] use it for simplification. The effectiveness of these works and our experiments support this approximation.
>
> **Extensibility**: Our proof can be extended to other distributions. For example, suppose in eq. (17), $\delta_i \sim Q$, where Q is an unknown distribution with a mean of 0 (we will explain why this is reasonable). According to GMM[4], $p(\delta_i) = \sum_{j=1}^M \alpha_j N_j(\mu_j,\Sigma_j)$, where $\sum_{j=1}^M \alpha_j = 1$. This means our theorems are applicable to nearly all distributions. Details will be shown in the paper.
>
> Then we address why we can assume $\mu_{\delta} = 0$. Considering $P$ as a manifold, the internal noise of $P$ is scattered because each point within $P$ has varying similarity to other manifolds. The noise in all directions makes the assumption reasonable, indicating noise does not affect the manifold's centroid.
>
> In summary, our theory is sound and can be extended to nearly all noise distributions.
>
> We hope the revised version better meets your expectations, if you can improve the score then we appreciate it.
>
> **Reference:**
>
> [1] Yang, Xiaocheng, et al. "Simple and efficient heterogeneous graph neural network." *Proceedings of the AAAI conference on artificial intelligence*. Vol. 37. No. 9. 2023.
>
> [2] Lee, Kuang-Huei, et al. "Cleannet: Transfer learning for scalable image classifier training with label noise." *Proceedings of the IEEE conference on computer vision and pattern recognition*. 2018.
>
> [3] Zhu, Xingquan, and Xindong Wu. "Class noise vs. attribute noise: A quantitative study." *Artificial intelligence review* 22 (2004): 177-210.
>
> [4] Reynolds, Douglas A. "Gaussian mixture models." *Encyclopedia of biometrics* 741.659-663 (2009).

---

### Official Review · Reviewer_nFdn · 2024-07-13

**Soundness:** 3
**Presentation:** 3
**Contribution:** 3
**Rating:** 7
**Confidence:** 4

**Summary:**

This paper aims to improve the performance of graph neural networks (GNNs) for node classification problem by generating high-quality labeled nodes and enriching node neighbors. It first uses pseudo-labeling to transform the unlabeled nodes into low-quality labeled nodes and performs mixup to generate high-quality labeled nodes. It then adopts an ensemble technique to establish an information exchange path between nodes of the same classes. Extensive experiments have been conducted to verify the proposed method.

**Strengths:**

1. The proposed method IntraMix could simultaneously generate high-quality labeled nodes and enrich node neighbors, and the experimental results on seven datasets show the superiority of the proposed method compared with other graph augmentation methods.

2. The authors describe the motivations of the intra-class mixup and neighbor selection (section 3.1 and 3.2) clearly, and provide the corresponding theoretical proofs to guarantee the effectiveness of the method.

3. Time complexity analysis is discussed (section 3.4), which illustrates the complexity of IntraMix is in the same of order of magnitude as the traditional GNN.

4. Since IntraMix is a data augmentation framework, it is could be easily incorporated with existing GNNs.

**Weaknesses:**

1. In Table 1, the improvements of the node classification accuracy of the proposed method compared with other graph augmentation methods, such as NodeMixup and Local Augmentation, is not significant, which should be discussed more clearly.

2. Although Theorem 3.1 and 3.2 proof that the noise in the generated data is smaller than that in the original data, the useful information for node classification should not be reduced and the theoretical analysis of the generated data containing sufficient information for classification is needed.

**Questions:**

1. One of my concerns is that whether the proposed method might loss some useful information when eliminating the noise in the original graph? The proposed method should contain sufficient information for node classification and the corresponding analysis should be proofed or discussed.

2. Since the neighbor selection operation is to enrich the node neighbors, it is similar to graph structure learning (GSL) methods that learn optimal graph structure for GNNs. What is the advantage of the proposed method compared with GSL methods?

**Limitations:**

Please refer to weakness and questions.

---

> ### Author Rebuttal · Authors · 2024-08-06
>
> Thank you for your constructive comments and suggestions. Next is our response.
>
> **Q1. In Table 1, the improvements of the proposed method is not significant, which should be discussed more clearly.**
>
> A1. Thank you for raising this question. We will discuss it further.
>
> **Significance:** We believe that exploring new research directions impacts the field's development more than achieving high performance. **The most significant contribution of IntraMix is not in substantially improving performance but in uncovering the largely overlooked potential of Intra-Class Mixup, and providing ample analysis.** IntraMix effectively addresses the challenges of generating low-noise nodes and constructing their neighbors. It opens new possibilities for graph augmentation, which is of great importance for the development of graph domain.
>
> **Practicality:**  Regarding accuracy, unlike previous methods that perform poorly on some datasets, IntraMix is the only one that **maintains a stable advantage across most datasets. This means we can always trust IntraMix.** For larger improvements, we can replace the simple pseudo-labeling method or switch to more refined neighbor selection methods. **They can be easily integrated into IntraMix.**
>
> The superiority of a method is not only reflected in its accuracy but also in **its ease of use**. IntraMix excels here, requiring minimal workflow modifications and no complex learnable ways. Sec 3.4 highlights its low time complexity, making it practical for deployment. Minor adjustments in the graph learning pipeline can greatly improve performance with minimal additional cost, facilitating the method's application.
>
> In summary, IntraMix is highly meaningful not only for its stable performance improvement but also for introducing an overlooked way in graph augmentation. Additionally, it offers great advantages in terms of ease of use.
>
> **Q2. The generated data containing sufficient information for classification is needed.**
>
> A2. Thank you for raising this issue. We provide a brief analysis here. Details will be updated in the paper.
>
> According to [1], consider $\lambda\sim Beta(\alpha,\alpha),j \sim U(n),\alpha>0$. Our loss function can be represented as:
> $$
> l(f)=\frac{1}{n}\sum_{i=1}^nl(\hat y_i,f(\hat x_i))+R_1(f)+R_2(f)+R_3(f)+R_4(f),
> $$
> where
> $$
> R_1(f)=\frac{1}{2n}\sum_{i=1}^n||(\nabla f(\hat x_i)-J^{(i)})^T(\nabla^2_{uu}l(\hat y_i,f(\hat x_i)))^\frac{1}{2}||^2_{\Sigma_{\hat x\hat x}^i},
> $$
>
> $$
> R_2(f)=\frac{1}{2n}\sum_{i=1}^n<\Sigma_{\hat x \hat x}^i,{\nabla_ul(\hat y_i,f(\hat x_i))\nabla^2f(\hat x_i)}>,
> $$
>
> $$
> R_3(f)=-\frac{1}{2n}\sum_{i=1}^n||\Sigma_{\hat x\hat y}^i\nabla^2_{yu}l(\hat y_i,f(\hat x_i))(\nabla^2_{uu}l(\hat y_i,f(\hat x_i)))^{-\frac{1}{2}}||^2_{(\Sigma_{\hat x\hat x}^i)^{-1}},
> $$
>
> $$
> R_4(f)=\frac{1}{2n}\sum_{i=1}^n<\Sigma_{\hat y\hat y}^i,{\nabla^2_{yy}l(\hat y_i,f(\hat x_i))>}.
> $$
>
> Then, Intra-Class Mixup can be seen as better utilizing data by adding regularization terms. From this perspective, it does not lead to a loss of information.
>
> Moreover, many works [2] have shown that Mixup enhances the ability to utilize data rather than causing information loss. Based on these theoretical foundations and validated by the effectiveness of IntraMix in experiments, IntraMix does not lead to information loss.
>
> **Q3. The proposed method should contain sufficient information and the analysis should be proofed.**
>
> A3. Please refer to A2.
>
> **Q4. What is the advantage of the proposed method compared with GSL methods?**
>
> A4. Thank you for raising this question.
>
> Our biggest advantage over GSL methods lies in **training and deployment costs**. GSL methods require learnable ways for graph topologies, leading to high training costs (e.g., GAUG[3] uses GAE for edge probabilities, and Nodeformer[4] uses differentiable ways). In contrast, as analyzed in Sec 3.4, IntraMix has very low time costs, making it more practical for deployment.
>
> Secondly, we present comparative results with GSL methods.
>
> |            | Cora         | CiteSeer     | Pubmed       |
> | ---------- | ------------ | ------------ | ------------ |
> | GCN        | 81.51 ± 0.42 | 70.30 ± 0.54 | 79.06 ± 0.31 |
> | SE-GSL [5] | 82.52 ± 0.51 | 71.08 ± 0.76 | 80.11 ± 0.45 |
> | LAGCN      | 84.61 ± 0.57 | 74.70 ± 0.51 | 81.73 ± 0.71 |
> | IntraMix   | 85.25 ± 0.42 | 74.80 ± 0.46 | 82.98 ± 0.54 |
>
> We find that IntraMix outperforms current popular GSL methods due to its effective use of low-quality labels and efficient topology optimization.
>
> Next, we incorporate a GAE in IntraMix, as detailed in [3].
>
> |          | Cora       | CiteSeer   | Pubmed     |
> | -------- | ---------- | ---------- | ---------- |
> | IntraMix | 85.25±0.42 | 74.80±0.46 | 82.98±0.54 |
> | +GAE     | 85.99±0.30 | 75.19±0.33 | 83.52±0.27 |
>
> While this improves performance, it adds training costs for GAE, presenting a trade-off.
>
> In summary, IntraMix offers advantages over GSL methods with lower training and deployment costs and shows versatility by integrating various GSL methods, highlighting its potential for performance enhancement.
>
> We hope the revised version better meets your expectations, if you can improve our score then we appreciate it.
>
> **Reference:**
>
> [1] Carratino, Luigi, et al. "On mixup regularization." *Journal of Machine Learning Research* 23.325 (2022): 1-31.
>
> [2] Beckham, Christopher, et al. "On adversarial mixup resynthesis." *Advances in neural information processing systems* 32 (2019).
>
> [3] Zhao, Tong, et al. "Data augmentation for graph neural networks." *Proceedings of the aaai conference on artificial intelligence*. Vol. 35. No. 12. 2021.
>
> [4] Wu, Qitian, et al. "Nodeformer: A scalable graph structure learning transformer for node classification." *Advances in Neural Information Processing Systems* 35 (2022): 27387-27401.
>
> [5] Zou, Dongcheng, et al. "Se-gsl: A general and effective graph structure learning framework through structural entropy optimization." *Proceedings of the ACM Web Conference 2023*.

---

> > ### Comment · Reviewer_nFdn · 2024-08-11
> >
> > Thanks for your detailed feedback, which has addressed my concerns on the improvements in experiments and comparison with GSL methods. Thus, I will maintain my positive score.

---

> > > ### Author Response · Authors · 2024-08-13
> > >
> > > Thank you for your response. We are pleased that we were able to address your concerns. We also want to express our sincere gratitude once again for your selfless efforts during the review process.

---

### Official Review · Reviewer_PZ8r · 2024-07-13

**Soundness:** 2
**Presentation:** 3
**Contribution:** 2
**Rating:** 5
**Confidence:** 4

**Summary:**

This paper proposes an intra-class mixup generation method to generate high-quality labeled data to improve the performance in the node classification task.

**Strengths:**

1. The extensive experimental results show that the proposed IntraMixup outperforms most of the baseline methods across several datasets.
2. The authors provide the theoretical analysis to support the effectiveness of the proposed method.
3. The presentation of this paper is good and it's easy to follow.

**Weaknesses:**

1. I am not fully convinced by the effectiveness of IntraMix by only generating node based on pseudo-labeled nodes and labeled nodes. NodeMixup has both the intra-class mixup and inter-class mixup. NodeMixup has similar node selection method to generate nodes from the same class but the different neighbor selection criteria. Why does the proposed method achieves much better performance than NodeMixup? What's the advantage of the proposed IntraMix over NodeMixup?
2. The novelty of the proposed method is somehow limited as it is similar to NodeMixup.

**Questions:**

1. What are the values of $\eta_1$ and $\eta_2$ when the proposed method achieves the best performance on different datasets?
2. In the ablation study, by saying "replacing the generated nodes with all-zero vector" do you mean that the node feature is a all-zero vector?
3. NodeMixup has both the intra-class mixup and inter-class mixup. Why does the proposed method achieves much better performance than NodeMixup? What's the advantage of the proposed IntraMix over NodeMixup?
4. What's the performance of Vanilla Mixup on the heterophilic graphs? Does the proposed method has better performance than Vanilla Mixup?

**Limitations:**

The authors list the limitation of the proposed method in appendix D.

---

> ### Author Rebuttal · Authors · 2024-08-06
>
> Thank you for your constructive comments and suggestions.  Next is our response.
>
> **Q1.Why does the proposed method achieves better performance than NodeMixup?**
>
> A1. This is a great question. We discuss our advantages over NodeMixup in Sec 1, and Appendix E.2. We analyze the reasons in terms of accuracy and time complexity.
>
> **A1.1 Accuracy:** We discuss Mixup and neighbor selection separately.
>
> **Mixup:** According to Appendix B.1.2, Inter-Class Mixup cannot guarantee the noise reduction within each class, while NodeMixup, which primarily relys on Inter-Class Mixup, is more susceptible to label noise compared to IntraMix.
>
> **Neighbor Selection:** NodeMixup uses neighborhood similarity as a similarity metric, but this fails because the generated data lack neighbors. IntraMix avoids this by selecting same-class nodes, enabling a cold start on generated data. Additionally, NodeMixup's selected nodes are overly similar, resulting in less information gain. In contrast, neighbors selected by IntraMix may not have high feature similarity, leading to greater information gain.
>
> **A1.2 Time complexity:** As analyzed in Sec 3.4, our neighbor selection process has a time complexity of O(n) compared to NodeMixup's $O(n^2)$, where n is the number of augmented nodes. Thus, IntraMix has a time complexity advantage. We show experimental results below.
>
> |           | Pubmed(s) | ogbn-arxiv(5%)(s) |
> | --------- | --------- | ----------------- |
> | IntraMix  | 3.24±0.35 | 17.28±1.99        |
> | NodeMixup | 5.88±0.87 | 30.11±2.61        |
>
> As observed, NodeMixup is much more costly than IntraMix, demonstrating IntraMix's cost superiority and greater advantage for large graphs, proving its general applicability.
>
> **Q2. The novelty of the proposed method is somehow limited as it is similar to NodeMixup.**
>
> A2. We believe that IntraMix represents a significant innovation, rather than just a minor improvement of NodeMixup. Next, we will discuss this.
>
> **Intra-Class Miuxp:** The importance of IntraMix lies in its being **the first work to seamlessly integrate Mixup into node-level learning, providing detailed experimental and theoretical analysis**. Through IntraMix, we shift the perspective that Inter-Class Mixup is more important for node-level tasks. Previous works, including NodeMixup, primarily focused on Inter-Class Mixup. They involved a transplant of Mixup into the graph domain with little consideration of graph-specific characteristics. Through rigorous experiments and theorems, we show that the neglected Intra-Class Mixup, is indeed more suitable for node-level learning. **This is important for the graph field as it explores a more promising and theoretically supported data augmentation way**.
>
> **Neighborhood Selection:** The innovation of our neighbor selection lies in using an **efficient method to achieve superior results compared to previous computationally intensive ways**. Details can be found in Sec 3.2. We naturally leverages the characteristics of Intra-Class Mixup to address a major challenge in applying Mixup to graphs—neighbor selection for generated data—and provides rigorous theoretical and experimental validation. Previous neighbor selection schemes, including NodeMixup, require significant costs. IntraMix achieves superior results in a more efficient way. Therefore, we consider our neighborhood selection method to be highly innovative. Moreover, it **combines with Intra-Class Mixup to form a beautiful and effective data augmentation framework**.
>
> In summary, IntraMix exhibits significant innovation over NodeMixup. **Its elegant design, rigorous proof, and extensive experiments challenge previous understandings of node-level data augmentation**. This has important implications for graph machine learning and will inspire future research.
>
> **Q3. What are the values of $\eta$ when IntraMix achieves the best performance?**
>
> A3. This is a great question. We present the results on GIN and show the values of $\eta$ related to the best results, averaged over 10 runs.
>
> |          | $\eta_1$ | $\eta_2$ |
> | -------- | -------- | -------- |
> | Cora     | 1.8±0.1  | 1.7±0.1  |
> | CiteSeer | 2.0±0.2  | 2.0±0.1  |
> | Pubmed   | 1.3±0.4  | 1.1±0.4  |
>
> According to Theorem 3.2, a larger $\eta$ reduces noise better. However, since learnable $\eta$ also affects the GNN's learning capability, isolating the optimal values for these two tasks is difficult. Thus, observing its value in relation to noise reduction is not very meaningful.
>
> **Q4. By saying "replacing ... with all-zero vector" do you mean that the node feature is a all-zero vector?**
>
> A4. We apologize for any confusion caused. Your understanding is correct. We conduct such experiments, showing that IntraMix's improvement is due not only to optimal topology but also to the nodes generated through Mixup.
>
> **Q5. Does IntraMix have better performance than Vanilla Mixup on the heterophilic graphs?**
>
> A5. This is a great question. We conduct experiments on four heterophilic graphs.
>
> |                | Cornell  | Texas    | Wisconsin | Squirrel |
> | -------------- | -------- | -------- | --------- | -------- |
> | ACMGCN         | 85.1±6.0 | 89.1±3.0 | 86.6±4.6  | 55.1±1.5 |
> | +Vanilla Mixup | 84.5±3.4 | 87.9±4.2 | 82.8±4.1  | 54.2±1.3 |
> | +NodeMixup     | 84.9±2.4 | 88.8±3.9 | 84.9±3.8  | 54.8±1.3 |
> | +IntraMix      | 87.2±3.6 | 90.3±2.8 | 89.4±4.0  | 59.4±1.3 |
>
> We find that IntraMix achieves stable improvements on heterophilic graphs, unlike Vanilla Mixup and NodeMixup, which degrade in performance. This is due to the critical role of neighbor selection in heterophilic data, where Vanilla Mixup and NodeMixup fail to identify suitable neighbors, leading to impacting performance. IntraMix contributes to information gain, as detailed in Appendix D.2, thus demonstrating its advantage on heterophilic graphs and general applicability.
>
> We hope the response meets your expectations, if you can improve the score then we appreciate it.

---

> > ### Comment · Reviewer_PZ8r · 2024-08-11
> > **Reply to authors' rebuttal**
> >
> > I thank the authors for their thoughtful rebuttal. My concerns have been properly addressed and I will maintain my positive score.

---

> > > ### Author Response · Authors · 2024-08-11
> > >
> > > We sincerely appreciate your response and the time you dedicated to reviewing our paper. Your valuable suggestions have been immensely helpful, and we will incorporate these into the revised version of our paper. We are also pleased to have addressed the concerns you raised about our work. If you would be willing to consider a potential increase in our score, we would be most grateful. Thank you once again for your support and guidance.

---

### Official Review · Reviewer_4UTx · 2024-07-18

**Soundness:** 3
**Presentation:** 2
**Contribution:** 2
**Rating:** 5
**Confidence:** 4

**Summary:**

This paper propose IntraMix, a data augmentation approach for node classification with graph neural networks. IntraMix effectively mixes node features of nodes in the same class based on pseudo labels to generate new nodes, and then link the generated nodes to selected nodes in the graph. The authors conduct some mathematical analysis on the method and demonstrate its effectiveness empirically.

**Strengths:**

- The proposed data augmentation approach is interesting.

- This paper conduct experiments on many datasets to justify the effectiveness of the proposed data augmentation approach. The authors report both accuracies and error bars in the results.

- Sec. 4.5 & 4.6 additionally assess the over-smoothing problem and performance on heterophilic graphs, broadening the scope of the study.

**Weaknesses:**

1. The statement of theoretical results looks problematic.
- Theorem 3.1:
  - Note that $P_{noise}(\cdot | x)$ and $P(\cdot | x)$ are both probabilistic distributions. Thus, the noise satisfies $\epsilon_1+\cdots+\epsilon_{|C|}=0$. The authors should clarify the distribution assumption on noises, since the constraint cannot be satisfied if $\epsilon_1,\cdots,\epsilon_{|C|}$ are i.i.d. Gaussian random variables. Besides, the proof seems to have overlooked this fact.
  - As a formal theorem statement, Theorem 3.1 should state line 128 using math equations for clarity.
  - I encourage the authors to discuss how $\lambda$ should be chosen based on this theorem.
- Theorem 3.2
  - The first sentence should be clearly stated as an assumption.
  - I'm suspicious about the correctness of the theorem. For example, if one sets $\eta_1=\eta_2=-\frac{1}{8(1+\lambda^2+(1-\lambda)^2)}-1$, then $(\lambda^2+(1-\lambda)^2)+\frac{1}{4(2+\eta_1+\eta_2)}=-1$. Thus, the formula in line 164 is wrong.

2. Some details of the experiment seem missing in the paper.
- I believe that semi-supervised learning is the most standard setting for node classification. The authors provide no reference on the inductive learning setting and supervised learning setting. It's not clear what those settings are and why they are important.
- The method relies on pseudo labels. The experiment section needs to state the way that the pseudo labels are generated and the cost of doing so.

3. Other issues.
- Line 91: Current definition implies that $|D_l|=|D_u|$, which is not general enough.
- Line 146: dropout rates $\to$ dropout probabilities.
- Line 169: lines 1 $\to$ line 1.

**Questions:**

- Does Eq. (5) mean that for any $(\hat x, y)\in D_m$ and any $(x_i, y)\in D_m$, $\hat x$ and $x_i$ are connected? If no, Eq. (5) should have been written differently, and the authors need to state how to select $(\hat x, y)\in D_m$ and any $(x_i, y)\in D_m$ and generate new edges.

**Limitations:**

The authors discuss the limitation of this paper in Appendix D.

---

> ### Author Rebuttal · Authors · 2024-08-06
>
> Thank you for your constructive comments and suggestions. Next is our response.
>
> **Q1. The authors should clarify the distribution assumption on noises, since the constraint cannot be satisfied if $\epsilon_1,...\epsilon_{|C|}$ are i.i.d. Gaussian random variables.**
>
> A1. We apologize if our expression caused any confusion. We believe this issue does not exist. Firstly, $\epsilon_1+...\epsilon_{|C|}=0$ may not hold. Here, $\epsilon_{i}$ represents the noise when a node is classified into class i, and this noise varies across different classes. From eq. (17), We can derive that without other constraints, the sum is not necessarily 0. Secondly, $\epsilon_1, \ldots, \epsilon_{|C|}$ are not independent due to class relationships, and whether their sum is 0 does not affect our analysis. We will update details in our paper.
>
> **Q2. Theorem 3.1 should state line 128 using math equations for clarity.**
>
> A2. Thank you for pointing out this issue. We can express it as *$P(\hat \epsilon<\epsilon)=\frac{2}{\pi} \arctan((\lambda^2+(1-\lambda)^2)^{-\frac{1}{2}})$, where $\hat \epsilon$ and $\epsilon$ represent the noise in the generated data and the original data, respectively*.
>
> **Q3. Discuss how λ should be chosen based on theorem 3.1.**
>
> A3. This is a great question. In the ablation study (line 290-294), we discussed this. The result is that when $\lambda=0.5$, IntraMix performs best, which aligns with Theorem 3.1. Similar to Mixup, to introduce randomness for better results, we set $\lambda$~B(2,2), where B denotes the Beta Distribution. $\lambda$ is around 0.5. Next, we provide more results.
>
> |                  | CiteSeer   | Pubmed     |
> | ---------------- | ---------- | ---------- |
> | $\lambda$=0.0    | 73.40±1.08 | 81.56±0.72 |
> | $\lambda$=0.1    | 74.08±0.69 | 82.04±0.59 |
> | $\lambda$=0.3    | 74.28±0.47 | 82.39±0.52 |
> | $\lambda$=0.5    | 74.58±0.59 | 82.77±0.49 |
> | $\lambda$~B(2,2) | 74.80±0.46 | 82.98±0.54 |
>
> It can be observed that the best results are achieved when $\lambda$ is around 0.5, which aligns with our analysis.
>
> **Q4.Theorem 3.2: The first sentence should be clearly stated as an assumption.**
>
> A4. Thank you for pointing out our imprecise expression. We will correct it.
>
> **Q5. I'm suspicious about the correctness of the theorem 3.2. For example, if $\eta_1=\eta_2=-\frac{1}{8(1+\lambda^2+(1-\lambda)^2)}-1$, the formula in line 164 is wrong.**
>
> A5. This is a great question, but $\eta_1$ and $\eta_2$ are learnable parameters, and $\eta$ is typically positive. If $\eta<0$,  it would imply that a node's own features negatively impact its classification, which is illogical. Such a situation would likely indicate issues with the dataset or training. Since $\eta<0$ does not typically occur, the values you mentioned would not arise.
>
> **Q6. It's not clear what inductive learning setting and supervised learning setting are and why they are important.**
>
> A6. Thank you for identifying this issue. Next, we will introduce them.
>
> **Inductive learning.** In node-level tasks, the common setting is transductive, where the test distribution is known during training, fitting many static graphs. Inductive learning refers to not knowing the test distribution during training. Since many real-world graphs are dynamic, inductive learning is also crucial. For inductive learning settings, we use the subgraph composed of training data for training, as referenced in Section 4.3 of [1].
>
> **Supervised learning.** Although graphs typically adhere to semi-supervised settings, some graphs, like citation networks, have enough labels. Therefore, we conduct supervised experiments to show that IntraMix is feasible not only in scenarios with scarce labels but also in those with sufficient labels, proving its generality. The setting involves using enough training labels, with details provided in Appendix C.1.
>
> **Q7. The experiment section needs to state the way that the pseudo labels are generated and the cost of doing so.**
>
> A7. Thank you for identifying this issue. The pseudo-labeling method involves training a GNN $f$ with scarce labeled data and using $f$ to get pseudo labels for unlabeled data, as referenced in [2]. We further train $f$ with the augmented data. The additional cost for pseudo-labeling is small; overall, we train only one model, and the combined cost of pre-training and training with augmented data does not greatly exceed normal training cost. We analyze the time complexity in Sec 3.4, and will update a detailed discussion.
>
> **Q8. Line 91: Current definition implies that $|D_l|=|D_u|$, which is not general enough.**
>
> A8. We apologize for the confusion caused by our expression; we will revise it in our paper. Typically, $|D_l| << |D_u|$, where $D_l$ denotes the set of labeled nodes and $D_u$ is the unlabeled node set.
>
> **Q9. Line 146: dropout rates → dropout probabilities；Line 169: lines 1 → line 1.**
>
> A9. Thank you for pointing out these issues. We will correct them.
>
> **Q10. Does Eq. (5) mean that for any $(\hat x,y)∈D_m$ and any $(\hat x_i,y)∈D_m$, $\hat x$ and $x_i$ are connected?  And the authors need to state how to select and generate new edges.**
>
> A10. We apologize for the confusion caused by our expression. We establish edges between $(\hat x, y) \in D_m$, and $(x, y) \in D_h$. Here, $D_m$ is the set of data generated by IntraMix. $D_h$ is the high-quality pseudo-label set, with the generation way in Eq. (4). For each node in $D_m$, we randomly choose some nodes from $D_h$ with the same label as neighbors by directly connecting them.
>
> We hope the revised version better meets your expectations, if you can improve the score then we appreciate it.
>
> **Reference:**
>
> [1] Liu, Songtao, et al. "Local augmentation for graph neural networks." *International conference on machine learning*. PMLR, 2022.
>
> [2] Lee, Dong-Hyun. "Pseudo-label: The simple and efficient semi-supervised learning method for deep neural networks." *Workshop on challenges in representation learning, ICML*. Vol. 3. No. 2. 2013.

---

> ### Comment · Reviewer_4UTx · 2024-08-10
>
> Thank you for the response. I will maintain my positive rating.
>
> Regarding Q5, I believe the correct way to state a theorem is to list $\eta\geq 0$ as an assumption and present experimental results to show this assumption holds in practice, otherwise the theorem itself is not mathematically correct.
>
> I encourage the authors to include the discussions in the paper for better clarity.

---

> > ### Author Response · Authors · 2024-08-11
> > **Rebuttal by Authors**
> >
> > Thank you for your response. We also greatly appreciate your selfless effort during the review process, pointing out the unclear expressions and areas that needed more detailed discussion. We will incorporate detailed discussions of these aspects in the revision of our paper to improve its quality.
> >
> > **Here, we provide a more detailed explanation of Q5**. We will explain why $\eta \geq 0$ from the perspectives of the meaning of $\eta$ and the experimental results.
> >
> > First, let's analyze the role of $\eta$. As described in [1], $\eta$ represents the extent to which the original features of the current node influence its feature embedding after learning through the GNN. Since the embedding representation of each node used in downstream tasks is derived from both its original features and the features of its neighbors, we generally believe that these two factors complement each other, with the current node's features playing a dominant role. Therefore, $\eta \geq 0$ is consistent with practical expectations. If $\eta < 0$, it would imply that, in generating the embedding representation of the current node, the neighborhood information is more important than the node's own features, which is unreasonable. For example, consider the task of node classification in a social network, where each node represents a user. Assuming that data collection is accurate and the machine learning model is properly trained, each user's own features should play the most significant role in their embedding representation, while the information from their friends serves only as an auxiliary factor. Therefore, $\eta \geq 0$ aligns with the practical scenario. Moreover, as long as $\eta > -1$, the noise reduction effect described in Theorem 3.2 holds. Thus, Theorem 3.2 is reasonable for practical use.
> >
> > Second, we validate the condition $\eta \geq 0$ from an experimental perspective. We present the results on GIN [1]. The table below shows the values of $\eta$ corresponding to the best experimental results when $\lambda=0.5$, averaged over 10 runs.
> >
> > |          | $\eta_1$  | $\eta_2$  |
> > | -------- | --------- | --------- |
> > | Cora     | 1.79±0.13 | 1.72±0.09 |
> > | CiteSeer | 2.03±0.15 | 1.96±0.16 |
> > | Pubmed   | 1.26±0.44 | 1.14±0.40 |
> >
> > From the experimental results, we can observe that $\eta \geq 0$ holds true. We will provide a more detailed discussion in the revised version of the paper.
> >
> > **Our response is as outlined above. We hope the revised version better meets your expectations, and we would greatly appreciate it if you could consider improving our score. If you have any further questions or suggestions, please feel free to reach out at any time. Thank you again.**
> >
> > **Reference:**
> >
> > [1] Xu, Keyulu, et al. "How Powerful are Graph Neural Networks?." *International Conference on Learning Representations*. 2019.

---

### Author Rebuttal · Authors · 2024-08-06

We thank all reviewers for their diligent efforts in evaluating our submission. We will revise the paper to address the comments raised by each reviewer. We are glad the reviewers find that

- Our method is elegant and interesting, efficiently addressing two major challenges in graph data augmentation: data generation and topology optimization. [4UTx, nFdn, bPiP]
- We use ample experiments to demonstrate the effectiveness of our method and its generality across different types of data. [4UTx, PZ8r, nFdn, bPiP]
- We provide thorough theoretical analysis, enhancing the credibility. [PZ8r, nFdn, bPiP]
- Our proposed method can be easily integrated into the GNN framework. [nFdn, bPiP]
- Our paper is well-written. [PZ8r, nFdn]

Next, we summarize the contributions of our work.

Our proposed IntraMix is the **first to discover and explain the significant potential of Intra-Class Mixup in graph representation learning**. We correct the excessive and inappropriate focus on Inter-Class Mixup in previous research, which is a strategy that applies Mixup to the graph domain lacked consideration of the characteristics of the graph domain. In constract, IntraMix leverages the characteristics of the graph domain more effectively, addressing the two main challenges in graph data augmentation: the low quality of generated nodes and the difficulty of selecting neighbors for these nodes. Additionally, we provide ample theoretical support. With these enhancements, we are confident that our paper will make an impactful contribution to the NeurIPS conference and the graph machine learning field.

Next, we will summarize some common issues. Due to space constraints, we cannot discuss these in detail here. **All questions raised by the reviewers will be discussed in detail in the revised paper.**

**Q1. The significance and novelty of our proposed method.**

A1. We believe that correcting the misplaced focus of previous research and proposing new research directions with solid theoretical support is more meaningful than merely improving model performance. We find that almost all existing works involving Mixup in the field of graph data augmentation focus on Inter-Class Mixup. This method does not take into account the characteristics of the graph domain, making it difficult to solve the problem of neighbor selection for generated data. We have discussed this in detail in Sec 1 and Sec 3.1. Through extensive theoretical and experimental validation, we demonstrate that due to the unique characteristics of the graph domain, the previously overlooked Intra-Class Mixup is more effective for node-level learning. We also introduce an efficient neighborhood selection method, forming an elegant IntraMix framework.

**Overall, the significance and novelty of IntraMix lies in shifting the focus from the inappropriate emphasis on Inter-Class Mixup to a more reasonable and theoretically sound approach for graph data augmentation.** We believe this will contribute to the development of graph machine learning and inspire future work. Further performance improvements can be achieved by replacing the simple pseudo-labeling method in IntraMix with more refined techniques and making minor adjustments to the neighbor selection. These could serve as future research directions.

**Q2. The deployment complexity and time complexity of our proposed method.**

A2. **Deployment complexity.** As described in Alg 1, IntraMix can be integrated into any GNN pipeline with very simple modification without involving any additional models that need training. Therefore, the overall deployment complexity is low, making it highly advantageous for practical use.

**Time complexity.** In Sec 3.4, we analyze the overall time complexity, which shows only a slight increase compared to a pipeline without data augmentation. Since only one model is trained overall, the additional training cost is minimal. The pseudo-labeling process involves only a small amount of model inference, the Mixup involves lightweight feature mixing without any learnable methods, and the neighbor selection process also involves minimal model inference. Overall, the time complexity of IntraMix is very low, making it practical for real-world applications.

In summary, IntraMix has low deployment and time complexity, making it very practical for real-world applications. We believe this is a highly promising method.

**Q3. The generalizability of our proof.**

A3. In eq. (17), we express the relationship between noise and feature distribution as: $P_{noise}(x|y_i)=P(x|y_i)+\delta_i$, where $\delta_i$ represents the noise. In the proofs of Theorem 3.1 and Theorem 3.2, we assume that the noise follows a Gaussian distribution.

First, this assumption is reasonable and widely adopted in current research. The Gaussian distribution is the most common noise distribution in reality, covering most scenarios. The effectiveness of previous works and our experimental results also support the reasonableness of this assumption.

Second, our proof can be extended to other distributions. For example, suppose in eq. (17), $\delta_i \sim Q$, where Q is an unknown distribution. According to GMM [1], $p(\delta_i) = \sum_{j=1}^M \alpha_j N_j(\mu_j,\Sigma_j)$, where $\sum_{j=1}^M \alpha_j = 1$. This means our theorems are applicable to nearly all distributions. We will provide a more detailed discussion in the paper.

We sincerely appreciate the reviewers' invaluable feedback and insightful suggestions.

**Reference:**

[1] Reynolds, Douglas A. "Gaussian mixture models." *Encyclopedia of biometrics* 741.659-663 (2009).

---

### Comment · Area_Chair_ymPM · 2024-08-11
**Dear reviewers, please read and respond to authors' rebuttal (if you haven't done so)**

Dear reviewers, please read and respond to authors' rebuttal (if you haven't done so). Thanks.

Your AC

---

### Decision · Program_Chairs · 2024-09-25

**Decision:**

Accept (poster)

**Comment:**

This submission received four ratings (5, 5, 7 and 5), averaging 5.5, which is above the borderline score. After rebuttal, three of four reviewers clearly expressed that their concerns have been well addressed. After carefully checking the concerns of the remaining silent reviewer and the considerable effort of the authors during rebuttal, I suggest the acceptance. Hope the authors carefully incorporate the advice into the submission to improve the final version.